# BIDIRECTIONAL LEARNING FOR OFFLINE MODEL-BASED BIOLOGICAL SEQUENCE DESIGN

## ABSTRACT

Offline model-based optimization aims to maximize a black-box objective function with a static dataset of designs and their scores. In this paper, we focus on biological sequence design to maximize some sequence score. A recent approach employs bidirectional learning, combining a forward mapping for exploitation and a backward mapping for constraint, and it relies on the neural tangent kernel (NTK) of an infinitely wide network to build a proxy model. Though effective, the NTK cannot learn features because of its parametrization, and its use prevents the incorporation of powerful pre-trained Language Models (LMs) that can capture the rich biophysical information in millions of biological sequences. We adopt an alternative proxy model, adding a linear head to a pre-trained LM, and propose a linearization scheme. This yields a closed-form loss and also takes into account the biophysical information in the pre-trained LM. In addition, the forward mapping and the backward mapping play different roles and thus deserve different weights during sequence optimization. To achieve this, we train an auxiliary model and leverage its weak supervision signal via a bi-level optimization framework to effectively learn how to balance the two mappings. Further, by extending the framework, we develop the first learning rate adaptation module *Adaptive-$\eta$*, which is compatible with all gradient-based algorithms for offline model-based optimization. Experimental results on DNA/protein sequence design tasks verify the effectiveness of our algorithm. Our code is available here.

## 1 INTRODUCTION

Offline model-based optimization aims to maximize a black-box objective function with a static dataset of designs and their scores. This offline setting is realistic since in many real-world scenarios we do not have interactive access to the ground-truth evaluation. The design tasks of interest include material, aircraft, and biological sequence (Trabucco et al., 2021). In this paper, we focus on biological sequence design, including DNA sequence and protein sequence, with the goal of maximizing some specified property of these sequences.

A wide variety of methods have been proposed for biological sequence design, including evolutionary algorithms (Sinai et al., 2020; Ren et al., 2022), reinforcement learning methods (Angermueller et al., 2019), Bayesian optimization (Terayama et al., 2021), search/sampling using generative models (Brookes et al., 2019; Chan et al., 2021), and GFlowNets (Jain et al., 2022). Recently, gradient-based techniques have emerged as an effective alternative (Trabucco et al., 2021). These approaches first train a deep neural network (DNN) on the static dataset as a proxy and then obtain the new designs by directly performing gradient ascent steps on the existing designs. Such methods have been widely used in biological sequence design (Norn et al., 2021; Tischer et al., 2020; Linder & Seelig, 2020). One obstacle is the out-of-distribution issue, where the trained proxy model is inaccurate for the newly generated sequences.

To mitigate the out-of-distribution issue, recent work proposes regularization of the model (Trabucco et al., 2021; Yu et al., 2021; Fu & Levine, 2021) or the design itself (Chen et al., 2022). The first category focuses on training a better proxy by introducing inductive biases such as robustness (Yu et al., 2021). The second category introduces bidirectional learning (Chen et al., 2022), which consists of a forward mapping and a backward mapping, to optimize the design directly. Specifically, the backward mapping leverages the high-scoring design to predict the static dataset and vice versa for

the forward mapping, which distills the information of the static dataset into the high-scoring design. This approach achieves state-of-the-art performances on a variety of tasks. Though effective, the proposed bidirectional learning relies on the neural tangent kernel (NTK) of an infinite-width model to yield a closed-form loss, which is a key component of its successful operation. The NTK cannot learn features due to its parameterization (Yang & Hu, 2021) and thus the bidirectional learning cannot incorporate the wealth of biophysical information from Language Models (LMs) pre-trained over a vast corpus of unlabelled sequences (Elnaggar et al., 2021; Ji et al., 2021).

To solve this issue, we construct a proxy model by combining a finite-width pre-trained LM with an additional layer. We then linearize the resultant proxy model, inspired by the recent progress in deep linearization (Achille et al., 2021; Dukler et al., 2022). This scheme not only yields a closed-form loss but also exploits the rich biophysical information that has been distilled in the pre-trained LM. In addition, the forward mapping encourages exploitation in the sequence space and the backward mapping serves as a constraint to mitigate the out-of-distribution issue. It is important to maintain an appropriate balance between exploitation and constraint, and this can vary across design tasks as well as during the optimization process. We introduce a hyperparameter $\gamma$ to control the balance, and develop a bi-level optimization framework *Adaptive-$\gamma$*. In this framework, we train an auxiliary model and leverage its weak supervision signal to effectively update $\gamma$. To sum up, we propose ***BI**directional learning for model-based **B**iological sequence design* (**BIB**). Last but not least, since the offline nature prohibits standard cross-validation strategies for hyperparameter tuning, all gradient-based offline model-based algorithms preset the learning rate $\eta$. There is a danger of a poor selection, and to address this, we propose to extend *Adaptive-$\gamma$* to *Adaptive-$\eta$*, which effectively adapts the learning rate $\eta$ via the weak supervision signal from the trained auxiliary model. To the best of our knowledge, *Adaptive-$\eta$* is the first learning rate adaptation module for gradient-based algorithms on offline model-based optimization. Experiments on DNA and protein sequence design tasks verify the effectiveness of BIB and *Adaptive-$\eta$*.

To summarize, our contributions are three-fold:

- Instead of adopting the NTK, we propose to construct a proxy model by combining a pre-trained biological LM with an additional trainable layer. We then linearize the proxy model, leveraging the recent progress on deep linearization. This yields a closed-form loss computation in bidirectional learning and allows us to exploit the rich biophysical information distilled into the LM via pre-training over millions of biological sequences.

- We propose a bi-level optimization framework *Adaptive-$\gamma$* where we leverage weak signals from an auxiliary model to achieve a satisfactory trade-off between exploitation and constraint.

- We further extend this bi-level optimization framework to *Adaptive-$\eta$*. As the first learning rate tuning scheme in offline model-based optimization, *Adaptive-$\eta$* allows learning rate adaptation for any gradient-based algorithm.

## 2 PRELIMINARIES

### 2.1 OFFLINE MODEL-BASED OPTIMIZATION

Offline model-based optimization aims to find a design $\boldsymbol{X}$ to maximize some unknown objective $f(\boldsymbol{X})$. This can be formally written as,

$$\boldsymbol{X}^* = \arg\max_{\boldsymbol{X}} f(\boldsymbol{X}),\tag{1}$$

where we have access to a size-$N$ dataset $\mathcal{D} = \{(\boldsymbol{X}_1, y_1)\}, \cdots, \{(\boldsymbol{X}_N, y_N)\}$ with $\boldsymbol{X}_i$ representing a certain design and $y_i$ denoting the design score. In this paper, $\boldsymbol{X}_i$ represents a biological sequence design, including DNA and protein sequences, and $y_i$ represents a property of the biological sequence such as the fluorescence level of the green fluorescent protein (Sarkisyan et al., 2016).

### 2.2 BIOLOGICAL SEQUENCE REPRESENTATION

Following (Norn et al., 2021; Killoran et al., 2017; Linder & Seelig, 2021), we adopt the position-specific scoring matrix to represent a length-$L$ protein sequence as $\boldsymbol{X} \in \mathbb{R}^{L \times 20}$, where 20 represents 20 different kinds of amino acids. For a real-world protein sequence, $\boldsymbol{X}[\boldsymbol{l}, :]$ ($0 \leq l \leq L-1$) is a

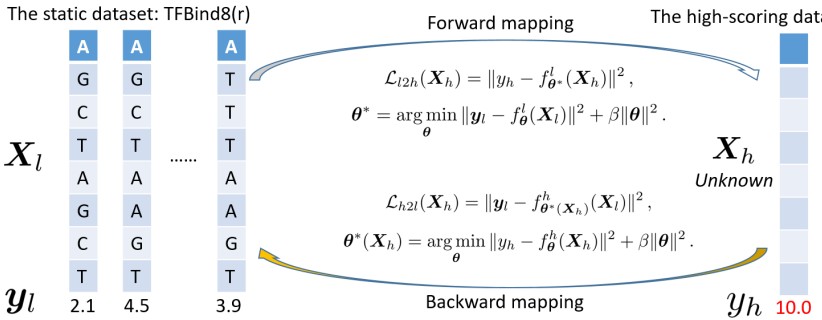

Figure 1: Illustration of bidirectional learning Chen et al. (2022) where $(\boldsymbol{X_l}, \boldsymbol{y_l})$ denotes the static dataset, $y_h$ is a large predefined target score and $\boldsymbol{X}_h$ is the high-scoring design we aim to find.

one-hot vector denoting one kind of amino acid. During optimization, $\boldsymbol{X}[\boldsymbol{l}, :]$ is a continuous vector and $softmax(\boldsymbol{X}[\boldsymbol{l}, :])$ represents the probability distribution of all 20 amino acids in the position $l$. Similarly, for a DNA sequence, we have $\boldsymbol{X} \in \mathbb{R}^{L \times 4}$ where 4 represents 4 different DNA bases.

The protein sequence $\boldsymbol{X}$ is fed into the embedding layer of the LM, which produces the embedding,

$$e = EMB(softmax(\boldsymbol{X})) \, . \tag{2}$$

The main block of the LM takes $e$ as input and outputs biophysical features. The DNA LM, which adopts the $k$-mer representation, is a little different from protein LMs. See Appendix A.1 for details.

## 2.3 Gradient ascent on sequence

A common approach to the posed offline model-based optimization problem is to train a proxy $f_{\boldsymbol{\theta}}(\boldsymbol{X})$ on the offline dataset,

$$\boldsymbol{\theta}^* = \arg\min_{\boldsymbol{\theta}} \frac{1}{N} \sum_{i=1}^{N} (f_{\boldsymbol{\theta}}(\boldsymbol{X}_i) - y_i)^2 \, . \tag{3}$$

Then we can obtain the high-scoring design $\boldsymbol{X}_h$ by $T$ gradient ascent steps:

$$\boldsymbol{X}_{t+1} = \boldsymbol{X}_t + \eta \nabla_{\boldsymbol{X}} f_{\boldsymbol{\theta}^*}(\boldsymbol{X})|_{\boldsymbol{X} = \boldsymbol{X}_t} \, , \quad \text{for } t \in [0, T-1] \, , \tag{4}$$

where the high-scoring design $\boldsymbol{X}_h$ can be obtained as $\boldsymbol{X}_T$.

Considering the discrete nature of biological sequences, the input of $f_{\boldsymbol{\theta}}(\cdot)$ should be discrete one-hot vectors. Following (Norn et al., 2021), we can perform the conversion and predict the score via:

$$\hat{\boldsymbol{X}}_i = softmax(\boldsymbol{X}_i) \, , \tag{5}$$

$$\boldsymbol{Z}_i = onehot(argmax(\hat{\boldsymbol{X}}_i)) \, , \tag{6}$$

$$\hat{y} = f_{\boldsymbol{\theta}}(\boldsymbol{Z}_i) \, . \tag{7}$$

Then the gradient regarding $\boldsymbol{X}_i$ can be approximated as,

$$\frac{df_{\boldsymbol{\theta}}(\boldsymbol{Z}_i)}{d\boldsymbol{x}_i} \approx \frac{df_{\boldsymbol{\theta}}(\boldsymbol{Z}_i)}{d\boldsymbol{z}_i} \frac{d\hat{\boldsymbol{x}}_i}{d\boldsymbol{x}_i} \, , \tag{8}$$

where we unroll the matrices $\boldsymbol{X}_i$, $\hat{\boldsymbol{X}}_i$ and $\boldsymbol{Z}_i$ as vectors $\boldsymbol{x}_i$, $\hat{\boldsymbol{x}}_i$ and $\boldsymbol{z}_i$ for notational convenience. This approximation allows us to use backpropagation directly from the proxy to the sequence design $\boldsymbol{X}_i$. For brevity, we will still use $f_{\boldsymbol{\theta}}(\boldsymbol{X_i})$ to represent the proxy.

## 2.4 Bidirectional learning

As shown in Figure 1, bidirectional learning (Chen et al., 2022), consists of two mappings: the forward mapping leverages the static dataset $(\boldsymbol{X_l}, \boldsymbol{y_l})$ to predict the score $y_h$ of the high-scoring

design $\boldsymbol{X}_h$, and the backward mapping leverages the high-scoring design data $(\boldsymbol{X}_h, y_h)$ to predict the static dataset $(\boldsymbol{X}_l, \boldsymbol{y}_l)$. The forward mapping loss can be written as

$$\mathcal{L}_{l2h}(\boldsymbol{X}_h) = \|y_h - f^l_{\boldsymbol{\theta}^*}(\boldsymbol{X}_h)\|^2 \,, \tag{9}$$

where $\boldsymbol{\theta}^*$ is given by

$$\boldsymbol{\theta}^* = \arg\min_{\boldsymbol{\theta}} \|\boldsymbol{y}_l - f^l_{\boldsymbol{\theta}}(\boldsymbol{X}_l)\|^2 + \beta\|\boldsymbol{\theta}\|^2 \,, \tag{10}$$

where $\beta > 0$ is a regularization parameter. The backward mapping loss can be written as

$$\mathcal{L}_{h2l}(\boldsymbol{X}_h) = \|\boldsymbol{y}_l - f^h_{\boldsymbol{\theta}^*(\boldsymbol{X}_h)}(\boldsymbol{X}_l)\|^2 \,, \tag{11}$$

where $\boldsymbol{\theta}^*(\boldsymbol{X}_h)$ is given by

$$\boldsymbol{\theta}^*(\boldsymbol{X}_h) = \arg\min_{\boldsymbol{\theta}} \|y_h - f^h_{\boldsymbol{\theta}}(\boldsymbol{X}_h)\|^2 + \beta\|\boldsymbol{\theta}\|^2 \,. \tag{12}$$

The high-scoring design $\boldsymbol{X}_h$ can be optimized by minimizing the bidirectional learning loss $\mathcal{L}(\boldsymbol{X}_h) = \mathcal{L}_{l2h}(\boldsymbol{X}_h) + \mathcal{L}_{h2l}(\boldsymbol{X}_h)$.

## 3 METHOD

In this section, we first illustrate how to leverage deep linearization to compute the bidirectional learning loss in a closed form. Subsequently, we introduce a hyperparameter $\gamma$ to control the balance between the forward mapping and the backward mapping. We then develop a novel bi-level optimization framework *Adaptive-$\gamma$*, which leverages a weak supervision signal from an auxiliary model to effectively update $\gamma$. Last but not least, we extend this framework to *Adaptive-$\eta$*, which enables us to adapt the learning rate $\eta$ for all gradient-based offline model-based algorithms. We summarize our method in Algorithm 1.

### 3.1 DEEP LINEARIZATION FOR BIDIRECTIONAL LEARNING

In bidirectional learning, the backward mapping loss is intractable for a finite neural network, so Chen et al. (2022) employ a neural network with infinite width, which yields a closed-form loss via the NTK. This however makes it impossible to incorporate the rich biophysical information that has been distilled into a pre-trained LM (Yang & Hu, 2021). Considering this, we construct a proxy model by combining a finite-width pre-trained LM with an additional layer. We then linearize the resultant proxy model, inspired by the recent progress in deep linearization which has established that an overparameterized DNN model is close to its linearization (Achille et al., 2021; Dukler et al., 2022).

Denote by $\boldsymbol{\theta_0} = (\boldsymbol{\theta_{pt}}, \boldsymbol{\theta}^{lin}_{init}) \in \mathcal{R}^{D \times 1}$ the proxy model parameters derived by combining the parameters of the pre-trained LM $\boldsymbol{\theta_{pt}}$ and a random initialization of the linear layer $\boldsymbol{\theta}^{lin}_{init}$. In this paper, we adopt the pre-trained DNABERT (Ji et al., 2021) and Prot-BERT (Elnaggar et al., 2021) models, and compute the average of token embeddings as the extracted feature, which is fed into the linear layer to build the proxy. Then we can construct a linear approximation for the proxy model:

$$f_{\boldsymbol{\theta}}(\boldsymbol{X}) \approx f_{\boldsymbol{\theta_0}}(\boldsymbol{X}) + \triangledown_{\boldsymbol{\theta}} f_{\boldsymbol{\theta_0}}(\boldsymbol{X}) \cdot (\boldsymbol{\theta} - \boldsymbol{\theta_0}) \,, \tag{13}$$

where $f_{\boldsymbol{\theta}}(\boldsymbol{X}), f_{\boldsymbol{\theta_0}}(\boldsymbol{X}) \in \mathcal{R}$, $\triangledown_{\boldsymbol{\theta}} f_{\boldsymbol{\theta_0}}(\boldsymbol{X}) \in \mathcal{R}^{1 \times D}$ and $\triangledown_{\boldsymbol{\theta}} f_{\boldsymbol{\theta_0}}(\boldsymbol{X}) \in \mathcal{R}^{D \times 1}$. Intuitively, if the fine-tuning does not significantly change $\boldsymbol{\theta_0}$, then this linearization is a good approximation. By leveraging this linearization, we can obtain a closed-form solution for Eq.(12) as:

$$\boldsymbol{\theta}^*(\boldsymbol{X}_h) = (\triangledown_{\boldsymbol{\theta}} f_{\boldsymbol{\theta_0}}(\boldsymbol{X}_h)^\top \triangledown_{\boldsymbol{\theta}} f_{\boldsymbol{\theta_0}}(\boldsymbol{X}_h) + \beta\boldsymbol{I})^{-1} \triangledown_{\boldsymbol{\theta}} f_{\boldsymbol{\theta_0}}(\boldsymbol{X}_h)\top(y_h - f_{\boldsymbol{\theta_0}}(\boldsymbol{X}_h)) + \boldsymbol{\theta_0}. \tag{14}$$

Building on this result, we can compute the bidirectional learning loss as:

$$\mathcal{L}_{bi}(\boldsymbol{X}_h) = \frac{1}{2}(\|y_h - \boldsymbol{K}_{\boldsymbol{X}_h \boldsymbol{X}_l}(\boldsymbol{K}_{\boldsymbol{X}_l \boldsymbol{X}_l} + \beta\boldsymbol{I})^{-1}(\boldsymbol{y}_l - f_{\boldsymbol{\theta_0}}(\boldsymbol{X}_l))\|^2$$
$$+ \|\boldsymbol{y}_l - \boldsymbol{K}_{\boldsymbol{X}_l \boldsymbol{X}_h}(\boldsymbol{K}_{\boldsymbol{X}_h \boldsymbol{X}_h} + \beta\boldsymbol{I})^{-1}(y_h - f_{\boldsymbol{\theta_0}}(\boldsymbol{X}_h))\|^2) \,, \tag{15}$$

where $\boldsymbol{K}(\boldsymbol{X}_i, \boldsymbol{X}_j) = \triangledown_{\boldsymbol{\theta}} f_{\boldsymbol{\theta_0}}(\boldsymbol{X}_i)\top \triangledown_{\boldsymbol{\theta}} f_{\boldsymbol{\theta_0}}(\boldsymbol{X}_j)$. Following (Dukler et al., 2022), we can also only linearize the last layer of the network for simplicity, which defines the following kernel,

$$\boldsymbol{K}(\boldsymbol{X}_i, \boldsymbol{X}_j) = BERT(\boldsymbol{X}_i)^\top BERT(\boldsymbol{X}_j), \tag{16}$$

where $BERT(\boldsymbol{X})$ denotes the feature of the sequence $\boldsymbol{X}$ extracted by BERT. Its kernel nature makes this approach suitable for small-data tasks (Arora et al., 2020), especially in drug discovery where the labeling cost of DNA/proteins is high.

---

**Algorithm 1** Bidirectional Learning for Offline Model-based Biological Sequence Design

---

    **Input:** Static dataset $\mathcal{D} = (\boldsymbol{X}_l, \boldsymbol{y}_l)$, predefined target score $y_h = 10$, # of iterations $T$,
    pre-trained biological LM parameterized by $\boldsymbol{\theta}_0$, auxiliary model $f_{aux}(\cdot)$, regularization $\beta$.
    **Output:** High-scoring design $\boldsymbol{X}_h^*$.
1: Initialize $\boldsymbol{X}_0$ as the sequence with the highest score in $\mathcal{D}$
2: **for** $\tau \leftarrow 0$ **to** $T - 1$ **do**
3:     Leverage Adaptive-$\gamma$ in Sec 3.2 to update the balance $\gamma$ by Eq. (21)
4:     **if** Adapt learning rate **then**
5:         Leverage Adaptive-$\eta$ in Sec 3.3 to update the learning rate $\eta$ by Eq. (23)
6:     Optimize $\boldsymbol{X}$ by minimizing the bidirectional learning loss $\mathcal{L}_{bi}(\boldsymbol{X}_\tau, \gamma)$ in Eq. (17):
7:         $\boldsymbol{X}_{\tau+1} = \boldsymbol{X}_\tau - \eta OPT(\nabla_{\boldsymbol{X}} \mathcal{L}_{bi}(\boldsymbol{X}_\tau, \gamma))$
8: Return $\boldsymbol{X}_h^* = \boldsymbol{X}_T$

---

## 3.2 ADAPTIVE-$\gamma$

The forward mapping and the backward mapping play different roles in the sequence optimization process: the forward mapping encourages the high-scoring sequence to search for a higher target score (exploitation) and the backward mapping serves as a constraint. Since different sequences require different degrees of constraint, we introduce an extra hyperparameter $\gamma \in [0, 1]$ to control the balance between the corresponding terms in the loss function:

$$\mathcal{L}_{bi}(\boldsymbol{X}_h, \gamma) = \gamma \mathcal{L}_{l2h}(\boldsymbol{X}_h) + (1 - \gamma)\mathcal{L}_{h2l}(\boldsymbol{X}_h). \tag{17}$$

Thus $\gamma = 1.0$ corresponds to the forward mapping alone, $\gamma = 0$ results in backward mapping, and $\gamma = 0.5$ leads to the bidirectional loss of (Chen et al., 2022).

It is non-trivial to determine the most suitable value for $\gamma$ since we do not know the ground-truth score for a new design. One possible solution is to train an auxiliary $f_{aux}(\cdot)$ to serve as a proxy evaluation. A reasonable auxiliary is a simple regression model fitted to the offline dataset. Although this auxiliary model cannot yield ground-truth scores, it can provide weak supervision signals to update $\gamma$, since the auxiliary model and the bidirectional learning provide complementary information. This is similar to co-teaching (Han et al., 2018) where two models leverage each other's view.

Formally, we introduce the *Adaptive-$\gamma$* framework. Given a good choice of $\gamma$, the produced $\boldsymbol{X}_h$ is expected to have a high score $f_{aux}(\boldsymbol{X}_h)$, based on which we can choose $\gamma$. To make the search for $\gamma$ more efficient, we can formulate this process as a bi-level optimization problem:

$$\gamma^* = \arg \max_\gamma f_{aux}(\boldsymbol{X}_h^*(\gamma)), \tag{18}$$

$$\text{s.t.} \quad \boldsymbol{X}_h^*(\gamma) = \arg \min_{\boldsymbol{X}_h} \mathcal{L}_{bi}(\boldsymbol{X}_h, \gamma). \tag{19}$$

We can then use the hyper-gradient $\frac{\partial f_{aux}(\boldsymbol{X}_h^*(\gamma))}{\partial \gamma}$ to update $\gamma$. Specifically, the inner level solution can be approximated via a gradient descent step with a learning rate $\eta$:

$$\boldsymbol{X}_h^*(\gamma) = \boldsymbol{X}_h - \eta \frac{d\mathcal{L}_{bi}(\boldsymbol{X}_h, \gamma)}{d\boldsymbol{X}_h^\top}. \tag{20}$$

For the outer level, we update $\gamma$ by hyper-gradient ascent:

$$\gamma = \gamma + \eta' \frac{df_{aux}(\boldsymbol{X}_h^*(\gamma))}{d\gamma} = \gamma + \eta' \frac{df_{aux}(\boldsymbol{X}_h)}{d\boldsymbol{x}_h} \frac{d\mathcal{L}_{bi}(\boldsymbol{X}_h, \gamma)}{d\boldsymbol{x}_h^\top}, \tag{21}$$

where we unroll the matrix $\boldsymbol{X}_h$ as a vector $\boldsymbol{x}_h$ for better illustration.

## 3.3 ADAPTIVE-$\eta$

We now extend the *Adaptive-$\gamma$* framework to *Adaptive-$\eta$*. As the first learning rate adaptation module for offline model-based optimization, *Adaptive-$\eta$* is compatible with all gradient-based algorithms and can effectively finetune the learning rate $\eta$ via the auxiliary model's weak supervision signal. All gradient-based methods that maximize $\mathcal{L}_{\boldsymbol{\theta}}(\boldsymbol{X})$ with respect to $\boldsymbol{X}$ have the following general form:

$$\boldsymbol{X}_{t+1} = \boldsymbol{X}_t + \eta OPT(\nabla_{\boldsymbol{X}} \mathcal{L}_{\boldsymbol{\theta}}(\boldsymbol{X})|_{\boldsymbol{X}=\boldsymbol{X}_t}), \quad \text{for } t \in [0, \mathrm{T} - 1], \tag{22}$$

where $\eta$ represents the learning rate of the optimizer. For methods such as simple gradient ascent (Grad), COMs (Trabucco et al., 2021), ROMA (Yu et al., 2021) and NEMO (Fu & Levine, 2021), $\mathcal{L}_{\boldsymbol{\theta}}(\cdot)$ is related to the proxy model $f_{\boldsymbol{\theta}}(\cdot)$; for BDI Chen et al. (2022) and our proposed method, BIB, $\mathcal{L}_{\boldsymbol{\theta}}(\cdot)$ is the negative of the bidirectional learning loss, i.e., $\mathcal{L}_{\boldsymbol{\theta}} = -\mathcal{L}_{bi}$.

Though the learning rate $\eta$ can be adapted in some optimizers such as Adam (Kingma & Ba, 2015), these adaptations rely on only the past optimization history and do not consider the weak supervision signal from the auxiliary model. Our *Adaptive-$\eta$* optimizes $\eta$ by solving:

$$\eta^* = \arg\max_{\eta} f_{aux}(\boldsymbol{X}_h^*(\eta)), \tag{23}$$

where $\eta$ can be updated via gradient ascent methods. Considering the sequence optimization procedure is highly sensitive to the learning rate $\eta$, we reset $\eta$ to $\eta_0$ at each iteration and update $\eta$ from $\eta_0$,

$$\eta = \eta_0 - \eta' \frac{df_{aux}(\boldsymbol{X}_h^*(\eta))}{d\eta}. \tag{24}$$

In general, this serves to stabilize the optimization procedure.

## 4 EXPERIMENTS

We conduct extensive experiments on DNA and protein design tasks, and aim to answer three research questions: (1) How does BIB compare with state-of-the-art algorithms? (2) Is every design component necessary in BIB? (3) Does the *Adaptive-$\eta$* module improve gradient-based methods?

### 4.1 BENCHMARK

We conduct experiments on two DNA tasks: TFBind8(r) and TFBind10(r), following (Chen et al., 2022) and three protein tasks: avGFP, AAV and E4B, in (Ren et al., 2022) which have the most data points. See See Appendix A.2 for details.

Following (Trabucco et al., 2021), we select the top $N = 128$ most promising sequences for each comparison method. Among these sequences, we report the maximum normalized ground truth score as the evaluation metric following (Ren et al., 2022).

### 4.2 COMPARISON METHODS

We compare BIB with two groups of baselines: the gradient-based methods and the non-gradient-based methods. For a fair comparison, the pre-trained LM is used for all methods involving a proxy and we don't finetune the LM. The gradient-based methods include: 1) Grad: gradient ascent on existing sequences to obtain new sequences; 2) COMs (Trabucco et al., 2021): lower bounds the DNN model by the ground-truth values and then applies gradient ascent; 3) ROMA (Yu et al., 2021): incorporates a smoothness prior into the DNN model before gradient ascent steps; 4) NEMO (Fu & Levine, 2021): leverages the normalized maximum-likelihood estimator to bound the distance between the DNN model and the ground-truth values; 5) BDI (Chen et al., 2022): adopts the infinitely wide neural network and its NTK to yield a closed-form bidirectional learning loss.

The non-gradient-based methods include: 1) BO-qEI (Wilson et al., 2017): builds an acquisition function for sequence exploration; 2) CMA-ES (Hansen, 2006): estimates the covariance matrix to adjust the sequence distribution towards the high-scoring region; 3) AdaLead (Sinai et al., 2020): performs a hill-climbing search on the proxy and then queries the sequences with high predictions; 4) CbAS (Brookes et al., 2019): builds a generative model for sequences above a property threshold and gradually adapts the distribution by increasing the threshold; 5) PEX (Ren et al., 2022): prioritizes the evolutionary search for protein sequences with low mutation counts; 6) GENH (Chan et al., 2021): enhances the score through a learned latent space.

### 4.3 TRAINING DETAILS

We follow the training setting in (Chen et al., 2022) if not specified. We choose $OPT$ as the Adam optimizer (Kingma & Ba, 2015) for all gradient-based methods. We implement the auxiliary model

Table 1: Experimental results (maximum normalized ground truth score) for comparison.

| Method | TFBind8(r) | TFBind10(r) | avGFP | AAV | E4B | Rank Mean | Rank Median |
|---|---|---|---|---|---|---|---|
| $\mathcal{D}$(**best**) | 0.242 | 0.248 | 0.742 | 0.452 | 0.224 | | |
| BO-qEI | $0.940 \pm 0.032$ | $0.595 \pm 0.028$ | $1.700 \pm 0.020$ | $\mathbf{0.591 \pm 0.002}$ | $0.436 \pm 0.004$ | 6.0/12 | 7.0/12 |
| CMA-ES | $0.930 \pm 0.034$ | $0.617 \pm 0.031$ | $5.488 \pm 0.056$ | $0.470 \pm 0.006$ | $0.748 \pm 0.009$ | 6.0/12 | 6.0/12 |
| AdaLead | $\underline{0.941 \pm 0.032}$ | $0.602 \pm 0.028$ | $1.611 \pm 0.009$ | $0.581 \pm 0.002$ | $0.433 \pm 0.003$ | 6.2/12 | 8.0/12 |
| CbAS | $0.878 \pm 0.049$ | $0.610 \pm 0.035$ | $1.371 \pm 0.016$ | $0.543 \pm 0.002$ | $0.349 \pm 0.003$ | 8.6/12 | 10.0/12 |
| PEX | $0.924 \pm 0.041$ | $0.612 \pm 0.026$ | $1.546 \pm 0.019$ | $\underline{0.588 \pm 0.002}$ | $0.397 \pm 0.004$ | 7.2/12 | 8.0/12 |
| GENH | $0.323 \pm 0.000$ | $0.448 \pm 0.000$ | $0.835 \pm 0.000$ | $0.452 \pm 0.000$ | $0.228 \pm 0.000$ | 11.4/12 | 11.0/12 |
| Grad | $\underline{0.941 \pm 0.026}$ | $0.630 \pm 0.029$ | $4.869 \pm 0.042$ | $0.463 \pm 0.005$ | $\underline{1.219 \pm 0.061}$ | 4.8/12 | 4.0/12 |
| COMs | $0.921 \pm 0.039$ | $0.637 \pm 0.065$ | $3.873 \pm 0.080$ | $0.511 \pm 0.005$ | $0.829 \pm 0.026$ | 5.6/12 | 5.0/12 |
| ROMA | $0.926 \pm 0.032$ | $0.634 \pm 0.061$ | $\underline{5.621 \pm 0.143}$ | $0.471 \pm 0.005$ | $1.198 \pm 0.042$ | 4.8/12 | 4.0/12 |
| NEMO | $0.930 \pm 0.038$ | $0.632 \pm 0.024$ | $4.624 \pm 0.087$ | $0.505 \pm 0.005$ | $1.036 \pm 0.046$ | 5.0/12 | 5.0/12 |
| BDI | $0.823 \pm 0.000$ | $\mathbf{0.678 \pm 0.000}$ | $0.742 \pm 0.000$ | $0.452 \pm 0.000$ | $0.224 \pm 0.000$ | 9.4/12 | 11.0/12 |
| **BIB**(ours) | $\mathbf{0.952 \pm 0.033}$ | $\underline{0.639 \pm 0.032}$ | $\mathbf{8.084 \pm 0.224}$ | $0.501 \pm 0.007$ | $\mathbf{1.255 \pm 0.029}$ | 2.4/12 | 1.0/12 |

as a linear layer with the feature from the pre-trained LM. We set the number of iterations $T$ as 25 for all experiments following (Norn et al., 2021) and $\eta_0$ as 0.1 following (Chen et al., 2022). We run every setting over 16 trials and report the average score. See Appendix A.3 for other details.

## 4.4 RESULTS AND ANALYSIS

We report all experimental results in Table 1 and plot the ranking statistics in Figure 2. We make the following observations. (1) As shown in Table 1, BIB consistently outperforms the Grad method on all tasks, which demonstrates that our BIB can effectively mitigate the out-of-distribution issue. (2) Furthermore, BIB outperforms BDI on 4 out of 5 tasks, which demonstrates the effectiveness of the pre-trained biological LM over NTK. The reason why BDI outperforms BIB on TFBind10(r) may be that short sequences do not rely much on the rich sequential information from the pre-trained LM. (3) As shown in Figure 2, the gradient-based methods generally perform better than the non-gradient-based methods, as also observed by Trabucco et al. (2021). (4) The gradient-based methods are inferior for the AAV task. One possible reason is that the design space of AAV ($20^{28}$) is much smaller than those of avGFP ($20^{239}$) and E4B ($20^{102}$), which makes the generative modeling and evolutionary algorithms more suitable. (5) This conjecture is also supported by the experimental results on two DNA design tasks. We compute the average ranking of gradient-based methods and non-grad-based methods on TFBind10(r) as 3.5 and 9.5, respectively, and the average ranking of gradient-based methods and non-grad-based methods on TFBind8(r) as 5.8 and 6.8, respectively. The advantage of gradient-based methods are larger ($9.5 - 3.5 = 6.0$) in TFBind10(r) than that ($6.8 - 5.8 = 1.0$) in TFBind8(r). (6) The generative modeling methods CbAS and GENH yield poor results on all tasks, probably because the high-dimensional data distribution is very hard to model. (7) Overall, BIB attains the best performance in 3 out of 5 tasks and achieves the best ranking results as shown in Table 1 and Figure 2.

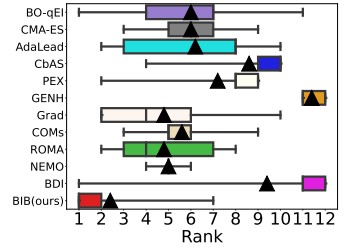

Figure 2: Rank minima and maxima are represented by whiskers; vertical lines and black triangles denote medians and means.

We also visualize the trend of performance (the maximum normalized ground truth score) and trade-off $\gamma$ as a function of $T$ on TFBind8(r) in Figure 3(a) and avGFP in Figure 3(b). The performance generally increases with the time step $T$ and then stabilizes, which demonstrates the effectiveness and robustness of BIB. Furthermore, we find that the $\gamma$ values of TFBind8(r) and avGFP generally increase at first. This means that BIB reduces the impact of the constraint to encourage a more aggressive search for a high target value during the initial phase. Then $\gamma$ of TFBind8(r) continues to increase while the $\gamma$ of avGFP decreases. We conjecture that the difference is caused by the sequence length. Small mutations of a biological sequence are enough to yield a good candidate (Ren et al., 2022). For the length-239 protein in avGFP, dramatic mutations 1) are not necessary and 2) can easily lead to out-of-distribution points. The weak supervision signal from the auxiliary model therefore encourages a tighter constraint towards the static dataset. By contrast, the DNA sequence is relatively short and a more widespread search of the sequence space can yield better results. To investigate this conjecture, we further visualize the trend of E4B in Figure 3(c). E4B also has long sequences (102) and we can observe its similar first-increase-then-decrease trend, although it is not as pronounced.

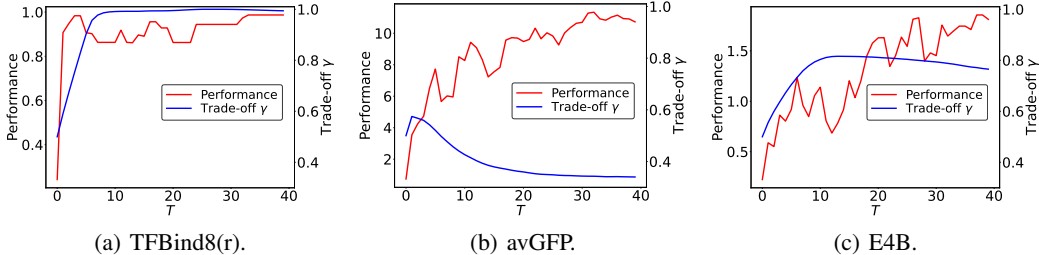

Figure 3: Trend of performance and trade-off $\gamma$ as a function of $T$.

Table 2: Ablation studies on BIB components.

| Task | $\gamma = 0.0$ | $\gamma = 1.0$ | $\gamma = 0.5$ | $\gamma = 0.5 + \text{Joint}$ | BIB | BIB + Ada-$\eta$ |
|---|---|---|---|---|---|---|
| TFBind8(r) | 0.936 | 0.933 | 0.947 | 0.935 | 0.952 | 0.956 |
| TFBind10(r) | 0.611 | 0.637 | 0.616 | 0.622 | 0.639 | 0.639 |
| avGFP | 6.051 | 7.588 | 7.940 | 7.920 | 8.084 | 8.197 |
| AAV | 0.449 | 0.458 | 0.480 | 0.420 | 0.501 | 0.525 |
| E4B | 0.778 | 0.903 | 1.198 | 1.176 | 1.255 | 1.301 |

## 4.5 ABLATION STUDIES

In this subsection, we conduct ablation studies to verify the effectiveness of the forward mapping, the backward mapping, and the *Adaptive-$\gamma$* module of BIB. We report the experimental results in Table 2.

**Forward mapping & Backward mapping.** We can observe that bidirectional learning ($\gamma = 0.5$) performs better than both forward mapping ($\gamma = 1.0$) and backward mapping ($\gamma = 0.0$) alone in most tasks, which demonstrates the effectiveness of forward mapping and backward mapping. The advantage of *bidirectional mappings over the forward mapping* is larger in the long-sequence tasks like avGFP (238) and E4B (102) compared with the short-sequence tasks. A possible explanation is that the constraint is more important for long sequence tasks than short sequence design since the search space is large and many mutations can easily go out of distribution.

**Adaptive-$\gamma$.** BIB learns $\gamma$ and this leads to improvements over bidirectional mappings ($\gamma = 0.5$) for all tasks, verifying the effectiveness of Adaptive-$\gamma$. We also consider the following variant,

$$\boldsymbol{X}^* = \arg \min_{\boldsymbol{X}_h} \mathcal{L}_{bi}(\boldsymbol{X}_h, 0.5) - f_{aux}(\boldsymbol{X}_h), \tag{25}$$

which jointly optimizes the bidirectional learning loss $\mathcal{L}_{bi}(\boldsymbol{X}_h, 0.5)$ and the auxiliary term $f_{aux}(\boldsymbol{X}_h)$. We found this yields similar or even worse results than pure bidirectional learning. The reason may be that the weak supervision signal from $f_{aux}(\boldsymbol{X}_h)$ can serve as a guide to update the scalar $\gamma$ but not as a component of the main optimization objective that directly updates the sequence.

In the final column of Table 2, we examine the performance of the Adaptive-$\eta$ module. Adding this module leads to improvements on all five tasks, which demonstrates its effectiveness.

## 4.6 ADAPTIVE-$\eta$

In this subsection, we aim to further demonstrate the effectiveness of the Adaptive-$\eta$ module on all six gradient-based methods. We conduct experiments on two tasks: TFBind8(r) and avGFP. Since the use of the infinitely wide neural network leads to poor performance for BDI, we modify its implementation via deep linearization so that it can make use of the pre-trained LM.

As shown in Table 3, *Adaptive-$\eta$* provides a consistent gain for all scenarios, which demonstrates the widespread applicability and effectiveness of the module. Furthermore, *Adaptive-$\eta$* leads to a maximum improvement of $1.4\%$ in TFBind8(r) and $12.5\%$ in avGFP. ROMA is the algorithm that benefits the most. One possible explanation is that ROMA incorporates a local smoothness prior that leads to more stable gradients, with which *Adaptive-$\eta$* can be more effective. Similar to Sec 4.5, we

Table 3: Adaptive-$\eta$ on all gradient-based methods.

| Method | TFBind8(r) | | | | | | avGFP | | | | | |
|---|---|---|---|---|---|---|---|---|---|---|---|---|
| | Grad | COMs | ROMA | NEMO | BDI | BIB | Grad | COMs | ROMA | NEMO | BDI | BIB |
| Normal | 0.941 | 0.921 | 0.926 | 0.930 | 0.947 | 0.952 | 4.869 | 3.873 | 5.621 | 4.624 | 7.940 | 8.084 |
| Joint | 0.941 | 0.921 | 0.931 | 0.932 | 0.935 | 0.925 | 4.869 | 3.836 | 6.438 | 3.078 | 7.920 | 7.823 |
| Gain | 0.000 | 0.000 | 0.005 | 0.002 | $-0.008$ | $-0.027$ | 0.000 | $-0.037$ | 0.116 | $-1.546$ | $-0.020$ | $-0.261$ |
| Ada-$\eta$ | 0.941 | 0.928 | 0.939 | 0.935 | 0.951 | 0.956 | 5.235 | 4.027 | 6.322 | 4.658 | 7.966 | 8.197 |
| Gain | 0.000 | 0.007 | 0.013 | 0.005 | 0.004 | 0.004 | 0.366 | 0.154 | 0.701 | 0.034 | 0.026 | 0.113 |

consider the following variant,

$$\boldsymbol{X}^* = \arg\max_{\boldsymbol{X}_h} \mathcal{L}_{\boldsymbol{\theta}}(\boldsymbol{X}_h) + f_{aux}(\boldsymbol{X}_h), \tag{26}$$

which performs joint optimization instead of bi-level optimization on two objectives. As shown in Table 3, joint optimization generally deteriorates the performance. This again verifies that the auxiliary model can only serve as a guide instead of contributing to the main objective.

## 5 RELATED WORK

**Biological sequence design.** There has been a wide range of algorithms for biological sequence design. Evolutionary algorithms (Sinai et al., 2020; Ren et al., 2022) leverage the learned surrogate model to provide evolution guidance towards the high-scoring region. Angermueller et al. (2019) propose a flexible reinforcement learning framework where sequence design is a sequential decision-making problem. Bayesian optimization methods propose candidate solutions via an acquisition function (Terayama et al., 2021). Deep generative model methods design sequences in the latent space (Chan et al., 2021) or gradually adapt the distribution towards the high-scoring region (Brookes et al., 2019). GFlowNets (Jain et al., 2022) amortize the cost of search over learning and encourage diversity. Gradient-based methods leverage a surrogate model and its gradient information to maximize the desired property (Chen et al., 2022; Norn et al., 2021; Tischer et al., 2020; Linder & Seelig, 2020). Our proposed BIB belongs to the last category and leverages the rich biophysical information (Ji et al., 2021; Elnaggar et al., 2021) to directly optimize the biological sequence.

**Offline model-based optimization.** A majority of sequence design algorithms (Angermueller et al., 2019; Sinai et al., 2020; Ren et al., 2022) focus on the online setting where wet-lab experimental results in the current round are analyzed to propose candidates in the next round. The problem of this setting is that wet-lab experiments are often very expensive, and thus a pure data-driven, offline approach is attractive and has received substantial research attention recently (Trabucco et al., 2022; Kolli et al., 2022). Gradient-based methods have proven to be effective (Trabucco et al., 2021; Yu et al., 2021; Fu & Levine, 2021; Chen et al., 2022). Among these algorithms, Chen et al. (2022) propose bidirectional mappings to distill information from the static dataset into a high-scoring design, which achieves state-of-the-art performances on a variety of tasks. However, this bidirectional learning is designed for general tasks, like robot and material design, and the rich biophysical information in millions of biological sequences is ignored. In this paper, we leverage recent advances in deep linearization to incorporate the rich biophysical information into bidirectional learning.

## 6 CONCLUSION

In this paper, we propose bidirectional learning for offline model-based biological sequence. Our work is built on the recently proposed bidirectional learning approach (Chen et al., 2022), which is designed for general inputs and relies on the NTK of an infinitely wide network to yield a closed-form loss computation. Though effective, the NTK cannot learn features. We build a proxy model using the pre-trained LM model with a linear head and apply the deep linearization scheme to the proxy, which can yield a closed-form loss and incorporate the wealth of biophysical information at the same time. In addition, we propose *Adaptive-$\gamma$* to maintain a proper balance between the forward mapping and the backward mapping by leveraging the weak supervision signal from an auxiliary model. Based on this framework, we further propose *Adaptive-$\eta$*, the first learning rate adaptation strategy compatible with all gradient-based offline model-based algorithms. Experimental results on DNA and protein sequence design tasks verify the effectiveness of BIB and *Adaptive-$\eta$*.

## 7 ETHICS STATEMENT

Protein sequence design aims to find a protein sequence with a particular biological function, which has a broad application scope. This can lead to improved drugs that are highly beneficial to society. For instance, designing the antibody protein for SARS-COV-2 can potentially save millions of human lives (Kumar et al., 2021) and designing novel anti-microbial peptides (short protein sequences) is central to tackling the growing public health risks caused by anti-microbial resistance (Murray et al., 2022). Unfortunately, it is possible to direct the research results towards harmful purposes such as the design of biochemical weapons. As researchers, we believe that we must be aware of the potential harm of any research outcomes, and carefully consider whether the possible benefits outweigh the risks of harmful consequences. We also must recognize that we cannot control how the research may be used. In the case of this paper, we are confident that there is a much greater chance that the research outcomes will have a beneficial effect. We do not consider that there are any immediate ethical concerns with the research endeavour.

## 8 REPRODUCIBILITY STATEMENT

We provide the code implementation of BIB and *Adaptive-η* here and we also attach the code in the supplementary material. We describe the DNA/protein benchmarks in Sec. 4.1 and the training details in Sec. 4.3. We also explain how to obtain the sequence embedding from the pre-trained LM and how to perform gradient ascent steps on the sequence in Sec. 2.

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

# A  APPENDIX

## A.1  DNA EMBEDDING

To incorporate richer contextual information, the DNA LM  Ji et al. (2021) adopts the $k$-mer sequence representation, which is widely used in DNA sequence analysis. For example, the sequence $ATGGCT$ has its 3-mer representation as $\{ATG, TGG, GGC, GCT\}$. In this paper, we adopt its 3-mer representation and compute the probability of the 3-mer token by multiplying the probabilities of the three individual bases. The 3-mer representation is then sent to the pre-trained DNA LM.

## A.2  DATASET DETAILS

We conduct experiments on two DNA tasks following (Chen et al., 2022) and three protein tasks in (Ren et al., 2022) which have the most data points. We report the dataset details in Table 4.

**DNA Task 1 TFBind8(r).** The goal is to find a length-8 DNA sequence to maximize the binding activity score with a particular transcription factor, SIX6REFR1 (Barrera et al., 2016). We sample 5000 data points for the offline algorithms following (Chen et al., 2022).

**DNA Task 2 TFBind10(r).** The task TFBind10(r) is the same as TFBind8(r) except that the goal is to find a length-10 DNA sequence. Both DNA tasks measure the entire search space and we adopt these measurements as the approximate ground-truth evaluation.

**Protein Task 1 avGFP.** This task aims to find a protein sequence with approximately 239 amino acids to maximize the fluorescence level of Green Fluorescent Proteins (Sarkisyan et al., 2016). The task oracle is constructed by using the full unobserved dataset (around 52,000 points) following (Ren et al., 2022). The oracle passes the average of the residue embeddings from the pre-trained Prot-T5 (Elnaggar et al., 2021) into a linear layer and then fits the dataset. The following two task oracles take the same form. The offline algorithms can only access the lowest-scoring 26,000 data points.

**Protein Task 2 AAV.** The goal is to engineer a 28-amino acid segment (positions 561–588) of the VP1 protein to remain viable for gene therapy (Bryant et al., 2021). We use the entire 284, 000 data points to build the oracle and the lowest-scoring 142, 000 points for the offline algorithms.

**Protein Task 3 E4B.** This task aims to design a protein (around 102 amino acids) to maximize the ubiquitination rate to the target protein (Starita et al., 2013). The full dataset consisting of around 100, 000 points is used to build the oracle and the bottom half is used for the offline algorithms.

The parameterization of the oracle is different from that of the regression model from two aspects: 1) model architecture; 2) pre-trained information source. First, the oracle adopts the Prot-T5 model

Table 4: Dataset details.

| Task | Metric | Min of $\mathcal{D}$ | Max of $\mathcal{D}$ | Min of $\mathcal{D}_{entire}$ | Max of $\mathcal{D}_{entire}$ |
|---|---|---|---|---|---|
| TFBind8(r) | binding activity | 0.000 | 0.242 | 0.000 | 1.000 |
| TFBind10(r) | binding activity | $-1.859$ | $-0.869$ | $-1.859$ | 2.129 |
| avGFP | fluorescence level | 1.283 | 3.553 | 1.283 | 4.123 |
| AAV | viruses viability | $-11.176$ | $-1.399$ | $-11.176$ | 9.536 |
| E4B | ubiquitination rate | $-3.589$ | $-0.984$ | $-3.589$ | 8.998 |

Table 5: Experimental results on different pre-trained LMs for comparison.

| Pre-trained LM | avGFP | AAV | E4B |
|---|---|---|---|
| ProtAlbert | $3.567 \pm 0.456$ | $0.478 \pm 0.004$ | $0.552 \pm 0.023$ |
| ProtBert$_{(adopted)}$ | $8.084 \pm 0.224$ | $0.501 \pm 0.007$ | $1.255 \pm 0.029$ |
| ProtBert-BFD | $8.240 \pm 0.094$ | $0.549 \pm 0.009$ | $1.880 \pm 0.054$ |

which consists of an encoder and a decoder, while the regression model adopts the Prot-BERT model which only has an encoder. Second, Prot-T5 is trained on the BFD and UniRef100 datasets and ProtBert is trained on the UniRef50 dataset. These two points demonstrate that the oracle and the regression model are different function classes. We choose the Prot-T5 model as the oracle because this is the state-of-the-art protein LM to extract features and recent work Elnaggar et al. (2021) has demonstrated its effectiveness. In order to test how related the Prot-T5 (oracle)/Prot-BERT(proxy) models are, we trained them on a sampled training dataset and compared the test predictions of the testing set. By evaluating the Pearson correlation coefficient (PCC) between the two prediction errors PCC(ProtT5 predictions - test labels, ProtBERT predictions -test labels), we obtain $-0.0053$ on avGFP, $-0.0005$ on AAV, and $-0.0062$ on E4B. These results suggest that the two models are not strongly related in terms of the predictions they form.

Following (Trabucco et al., 2021), we select the top $N = 128$ most promising sequences for each comparison method. Among these sequences, we report the maximum normalized ground truth score as the evaluation metric following (Ren et al., 2022).

## A.3 TRAINING DETAILS

We use Pytorch (Paszke et al., 2019) to run all experiments on one V100 GPU. Following the setting in Norn et al. (2021), we introduce a length-$L$ protein sequence as a continuous random matrix $\boldsymbol{X}_h \in R^{L \times 20}$ ($\boldsymbol{X}_h \in R^{L \times 4}$ for DNA), initialized using a normal distribution with the mean $0$ and the standard deviation of $0.01$. To make this sequence correspond correctly to the candidate sequence, we exchange the largest value in $\boldsymbol{X}[l, :]$ with the value in the amino acid index.

## A.4 DIFFERENT PRETRAINED LMS

As shown in Table 5, we have tested the ProtBERT, ProtAlbert, and ProtBert-BFD models and found that better-quality models generally work better. The publicly available pre-trained DNA models are limited and thus we only perform experiments on the protein tasks. Elnaggar et al. (2021) demonstrate that the language model performances follow the ordering: ProtBert-BFD > ProtBert > ProtAlbert. We can see that the performance ranks over the three protein tasks avGFP, AAV, and E4B are the same.

## A.5 DIFFERENT DATASET SIZE

As shown in Table 6, we have tested the performance of BDI as a function of dataset size (N= 20, 40, 60, 80, 100) in TFBind8(r) and TFBind10(r) since they have exact oracle evaluations. We see that performance is already good for N=20 for TFBind8(r) and N=40 for TFBind10(r).

## A.6 RANKING PERFORMANCE

As for prediction performances, the rank should be: a NN > linearized pre-trained LM > NTK. We have conducted experiments to verify this. We sample half of the data, train a model to predict another

Table 6: Experimental results on different size datasets for comparison.

| Dataset size | 20 | 40 | 60 | 80 | 100 |
|---|---|---|---|---|---|
| TFBind8(r) | $0.849 \pm 0.027$ | $0.883 \pm 0.036$ | $0.890 \pm 0.033$ | $0.911 \pm 0.042$ | $0.923 \pm 0.049$ |
| TFBind10(r) | $0.248 \pm 0.000$ | $0.596 \pm 0.035$ | $0.602 \pm 0.023$ | $0.616 \pm 0.024$ | $0.632 \pm 0.036$ |

Table 7: Mean squared prediction losses for comparison.

| Method | TFBind8(r) | TFBind10(r) | avGFP | AAV | E4B |
|---|---|---|---|---|---|
| Finetuned NN | $0.101 \pm 0.001$ | $1.130 \pm 0.041$ | $0.411 \pm 0.197$ | $5.148 \pm 0.074$ | $0.683 \pm 0.012$ |
| Linearized NN | $0.107 \pm 0.000$ | $1.618 \pm 0.000$ | $0.735 \pm 0.000$ | $23.041 \pm 0.000$ | $1.050 \pm 0.000$ |
| NTK | $0.111 \pm 0.000$ | $1.840 \pm 0.000$ | $0.807 \pm 0.000$ | $24.451 \pm 0.000$ | $1.075 \pm 0.000$ |

half data, and report the mean squared loss here and in Appendix A.6 Table 7. A small mean squared loss indicates a good prediction performance; thus, we have verified the above ranking order.

