# OpenReview forum: "Bidirectional Learning for Offline Model-based Biological Sequence Design"
_ICLR.cc/2023/Conference — Submitted to ICLR 2023_

### Official Review · Reviewer_Ay2G · 2022-10-20

**Confidence:** 4
**Correctness:** 4
**Technical Novelty And Significance:** 3
**Empirical Novelty And Significance:** 2
**Recommendation:** 5

**Clarity, Quality, Novelty And Reproducibility:**

The paper is largely experimental in nature by taking a previously published framework (Chen et al 2022) and applying a number of bells and whistles (bilevel optimization with adaptive hparam tuning, deep linearization, etc) to get better performance. However, lack of clarity in the exposition and training details makes the.
- The exposition would be improved with a diagram of the architecture setup for the Deep Linearization (and a general workflow diagram) as is common in DL papers.
- More care could be taken with the presentation of the equations to make the paper more readable.


**Strength And Weaknesses:**

Strengths
- The paper puts together an interesting set of ideas(bilevel optimization, deep linearization) that help resolve a weakness of prior work.
- The paper seems to perform better on benchmark for biological sequence design tasks.
- Adaptive hyperparameter tuning is extremely useful in model-based optimization settings due to the difficulty of tuning hyperparameters on out-of-distribution datasets.

Major Weaknesses
- Lack of clarity on major experimental details makes interpretation of results difficult: Is using the pretrained LM for all proxy methods shown to perform better than a standard CNN-type model as this has been shown to work well in FLIP (Dallago et al 2021)? Is there finetuning done on the LM used as a proxy for other methods? What is a "task oracle"? How were X_h and y_h initialized?
- The central premise of the paper is that the following rank order is expected for surrogate model models: a NN(perhaps finetuned version of pretrained LM ) which is intractable to optimize against > linearized pretrained LM > NTK. However, this is never shown in any ablation studies rather intuition about biophysical features are referenced. The paper would be strengthened a lot if it was shown that such a representation indicates a better surrogate model by demonstrating improved prediction performance(rather than optimization) on the biological sequence datasets shown later. This is because there is significant stochasticity in the ranking of the MBO methods depending on the experimental setting and hyperparameter choices so a better understanding of the central claim of the paper would


Minor Weaknesses:
- Confidence intervals should be provided for the scores.
- How were indels handled in the AAV task (given the biological sequence representation assumes fixed length $L$)?

**Summary Of The Paper:**

The paper tackles the model-based optimization problem for biological sequences. They extend the NTK-based bidirectional learning approach from previous work which essentially attempts to regularize according to features that transfer between low-fitness and high-fitness sequences. Their main contribution is twofold (1) they extend Chen et al 2022 by replacing the NTK representation with a pretrained language model with a linear layer atop and (2) they propose adaptive hyperparameter tuning strategies to improve the optimization scheme. They then benchmark their method against biological sequence datasets.

**Summary Of The Review:**

The paper has a nice set of ideas that could plausibly provide an improvement upon bidirectional learning proposed in (Chen et al 2022). However, the lack of care taken toward the presentation and experimental section in particular significantly hampers the ability of the paper to be published in its current form.

---

> ### Author Response · Authors · 2022-11-16
> **Response to review questions 1/n**
>
> ## General Reply
>
> We very much appreciate your effort in preparing such a careful review and we think that the modifications in response to your comments have improved the paper substantially. We provide responses to the individual comments below and explain how we have revised the paper to address them.
>
> ## Lack of clarity on major experimental details
>
> > Lack of clarity on major experimental details makes interpretation of results difficult: (1) Is using the pretrained LM for all proxy methods shown to perform better than a standard CNN-type model as this has been shown to work well in FLIP (Dallago et al 2021)? (2) Is there finetuning done on the LM used as a proxy for other methods? (3) What is a "task oracle"? (4) How were Xh and yh initialized?
>
> (1) It has been demonstrated in [1, 2] that the pre-trained LM is generally better than a standard CNN-type model. We conjecture that whether the adoption of pre-trained LM works well or not depends on the size of the training set. In FLIP, some of the datasets are large and thus the standard CNN-type model can achieve performance similar to that of the pre-trained LM. In this paper, we focus on biological sequence design and drug discovery where the data is usually scarce.
>
> (2) All methods, including ours, use an LM as part of the proxy. They do not fine-tune the LM and only use the extracted LM features. We have added a sentence ``For a fair comparison, the pre-trained LM is used for all methods involving a proxy and we do not fine-tune the LM.'' in Section 4.2.
>
> (3) When a method proposes some designs for a task, we need a task oracle to evaluate the quality of these newly proposed designs. The oracle is intended to provide something close to the ground-truth value.
> We build the task oracle as a regression model trained on a larger dataset following [1]. We have added text in the paper to clarify the meaning of a ``task oracle'' and specify how it is constructed.
>
> (4) $X_h$ is initialized as the best $128$ candidates in the offline dataset and $y_h$ is set as $10$ for all tasks following [2]. We have added text in the revised version to specify this.

---

> > ### Author Response · Authors · 2022-11-16
> > **Response to review questions 2/n**
> >
> > ## The central premise of the paper
> >
> > > The central premise of the paper is that the following rank order is expected for surrogate model models: a NN(perhaps finetuned version of pretrained LM ) which is intractable to optimize against > linearized pretrained LM > NTK. However, this is never shown in any ablation studies rather intuition about biophysical features are referenced. The paper would be strengthened a lot if it was shown that such a representation indicates a better surrogate model by demonstrating improved prediction performance (rather than optimization) on the biological sequence datasets shown later. This is because there is significant stochasticity in the ranking of the MBO methods depending on the experimental setting and hyperparameter choices so a better understanding of the central claim of the paper would
> >
> > Thanks for this valuable question and the suggestion to include results of this nature. We never claim ``a NN (perhaps finetuned version of pre-trained LM) which is intractable to optimize against > linearized pre-trained LM > NTK'' and we do not agree that this is the central premise of the paper. But the second relation (linearized pre-trained LM > NTK) is very important, and if this did not hold, our approach would be poorly motivated. We agree that additional experiments assessing predictive capability are a valuable addition.
> >
> > We agree that the proposed ranking should hold in the predictive setting.
> > To verify this, we have conducted further experiments. We sample half of the data, train a model to predict the metrics for the other half of the data, and report the mean squared loss. In Appendix A.6. a small mean squared loss means good prediction performance and thus we have verified the above ranking order. We observe that the linearized model achieves a $5$-$10$ percent improvement over the NTK for most tasks. The fine-tuned NN performs much better as a predictive model.
> >
> > |  Method  | TFBind8(r)  | TFBind10(r)   | avGFP | AAV | E4B |
> > |:----:|:----:|:----:|:----:|:----:|:----:|
> > Finetuned NN | ${0.101} \pm {0.001}$ | $1.130\pm {0.041}$ | $0.411\pm {0.197}$| $5.148\pm {0.074}$| $0.683\pm {0.012}$
> > Linearized NN | ${0.107} \pm {0.000}$ | $1.618\pm {0.000}$ | $0.735\pm {0.000}$| $23.041\pm {0.000}$| $1.050\pm {0.000}$
> > NTK | ${0.111} \pm {0.000}$ | $1.840\pm {0.000}$ | $0.807\pm {0.000}$| $24.451\pm {0.000}$| $1.075\pm {0.000}$
> >
> > In the context of bidirectional learning optimization, the ranking may differ. It has been proved in [2] that the NTK is better than a NN since the NN suffers from approximation errors caused by the gradient step solution, whereas the NTK yields a closed-form exact solution. When exposed to a smaller training set, with only low-scoring designs, there is the potential for the NN to overfit (even if we could efficiently optimize), and provide worse information in the out-of-distribution region that is being explored. By contrast, we have shown that the linearized LM can work with as few as 20 samples in the training set.
> >
> > The paper does rely on the premise that
> > the linearized pre-trained LM is superior to the NTK. Strictly speaking, this superiority is only necessary in the context of bidirectional learning for offline optimization, although as discussed above, we see that it also applies in the predictive setting. Our intuition is that the linearized pre-trained LM considers the biophysical features via the pre-trained LM and thus can incorporate valuable information learned from a much larger unlabelled corpus. From the experimental view, we run bidirectional learning on five tasks with (a) pre-trained LM and (b) NTK, and observe that (a) is better than (b) in five tasks (TFBind8: 0.947 $>$ 0.823; TFBind10: 0.616 $>$ 0.678; avGFP: 7.940 $>$ 0.742; AAV: 0.480 $>$ 0.452; E4B: 1.198 $>$ 0.224). The improvement is of a similar order to that observed in the predictive setting for most tasks. The exception is E4B, where the NTK performs unusually poorly.

---

> > > ### Author Response · Authors · 2022-11-16
> > > **Response to review questions 3/n**
> > >
> > > ## Minor Weaknesses
> > >
> > > > Confidence intervals should be provided for the scores.
> > >
> > > The $95$\% confidence interval of
> > > our method BIB over $16$ runs are $0.952 \pm 0.016$ on TFBind8(r), $0.639 \pm 0.016$ on TFBind10(r), $8.084 \pm 0.110$ on avGFP, $0.501 \pm 0.003$ on AAV and $1.255 \pm 0.014$ on E4B.  We have revised Table 1 and now also report the CIs of other methods in the main paper. We run every experiment 16 times and thus can estimate the 95\% confidence interval as 1.96*sigma/4. As we can see, the error bars are relatively small, which demonstrates the robustness of this setting.
> > >
> > > > How were indels handled in the AAV task (given the biological sequence representation assumes a fixed length
> > >
> > > Currently, we sidestep this issue
> > > by following the procedure in [3], which involves running multiple
> > > experiments by inputting sequences of different lengths (i.e. varying $L$).
> > > This allows us to generate candidate designs with different lengths but does not make the most effective use of the data.
> > >
> > > A possible alternative for handling indels in the optimization procedure would be to follow the approach in [4]. We can incorporate variable sequence lengths by maximizing the acquisition function over lengths $L-1$, $L$, and $L+1$ at each iteration. We then choose the largest as our new value of $L$ for the next iteration.
> > >
> > > ## Clarity, Quality, Novelty And Reproducibility
> > >
> > > > The exposition would be improved with a diagram of the architecture setup for the Deep Linearization (and a general workflow diagram) as is common in DL papers.
> > > More care could be taken with the presentation of the equations to make the paper more readable.
> > >
> > > Thank you for this suggestion. We have added a general workflow diagram Algorithm 1 in Section 3 of the main paper. We have also revisited every equation and made improvements to the presentation where necessary.
> > >
> > > ## Overall
> > >
> > > Does the above reply address your concerns? Thank you again for your instructive review and thoughtful feedback. We hope that there is the opportunity for further discussion with you during the rebuttal phase.
> > >
> > >
> > >     [1] Brandon Trabucco, Xinyang Geng, Aviral Kumar, and Sergey Levine. Design-Bench: benchmarks
> > >     for data-driven offline model-based optimization. arXiv preprint arXiv:2202.08450, 2022
> > >     [2] Can Chen, Yingxue Zhang, Jie Fu, Xue Liu, and Mark Coates. Bidirectional learning for offline
> > >     infinite-width model-based optimization. Proc. Adv. Neur. Inf. Proc. Syst (NeurIPS), 2022
> > >     [3] Christoffer Norn, Basile IM Wicky, David Juergens, Sirui Liu, David Kim, Doug Tischer, Brian Koepnick, Ivan Anishchenko, David Baker, and Sergey Ovchinnikov. Protein sequence design by conformational landscape optimization. Proceedings of the National Academy of Sciences, 2021.
> > >     [4] Yang Z, Milas K A, White A D. Now What Sequence? Pre-trained Ensembles for Bayesian Optimization of Protein Sequences[J]. bioRxiv, 2022.

---

> > > > ### Author Response · Authors · 2022-11-18
> > > > **Kindly Request for Feedback**
> > > >
> > > > Thanks for your insightful questions. We tried to thoroughly respond to your raised questions:
> > > >
> > > > - We have clarified major experimental details;
> > > > - We have illustrated the ranking order of the surrogate models;
> > > > - We have provided confidence intervals and explained how we handle indels;
> > > > - We have added a general workflow description (Algorithm 1) in Section 3 of the main paper to improve the exposition;
> > > > - We have revised the submission to address all your suggestions and concerns.
> > > >
> > > > Does the above reply address your concerns? Thank you again for your instructive review and thoughtful feedback.

---

### Official Review · Reviewer_bELc · 2022-10-23

**Confidence:** 3
**Correctness:** 3
**Technical Novelty And Significance:** 3
**Empirical Novelty And Significance:** 2
**Recommendation:** 5

**Clarity, Quality, Novelty And Reproducibility:**

Clarity: The method in the paper is clearly described, and I could follow the math. The experiments are not as clearly described. Some details are missing, and it's not clear what fitness metrics are being optimized.

Quality: The method is interesting and theoretically sound. It's difficult for me to judge the quality of the results due to the scale discrepancy described.

Novelty: I believe this method is novel, but I don't actively work in optimization and may not be aware of related work.

Reproducibility: Code is provided for the model. Reproducibility issues exist due to the scale discrepancy described.

**Strength And Weaknesses:**

Strengths
* New gradient based optimization scheme that can leverage pretrained language models
* Interesting approach for adapting the learning rate, which helps to deal with varying protein landscapes
* Math is correct and is fairly easy to follow
* Results appear relatively strong

Weaknesses
* Something is odd with the scale of the experiments. When I download the avGFP dataset, the maximum fitness value in the dataset is ~4.12, while the paper reports a value >8. The 4.12 number also appears consistent with the results in Ren et al. 2022. There also seems to be a scale mismatch in the AAV and E4B datasets between this paper and Ren et al. 2022 (in this paper, AAV fitness varies between 0.4 and 0.6, in Ren et al., it varies between -3.5 and 4.5; something similar is true for E4B).
    - It would be useful if the authors listed the exact fitness metric for each task and perhaps the maximum value achievable in the dataset. Additionally authors should explain or correct any scale mismatch when comparing with previous work.
    - This is the largest issue that must be addressed before publication - I would increase my score if this is addressed satisfactorily.
* The authors argue that the incorporation of pretrained language model features is a point in favor of their method. It would be interesting to see how the method performs as a function of language model quality. You could use different models from the Elnaggar et al. or the various sizes of the ESM language models and see if improvements in language modeling result in improvements in optimization, which would more directly show that these features have a significant effect on performance.
* Hard to follow what metrics are being reported in the table - could this be added to the table caption? It’s actually never explicitly said that the reported number is maximum fitness achieved, but I’m assuming it is
* It would be nice to motivate the problem setting better. I have a hard time understanding when this class of method would be useful. It purports to very quickly optimally sample a set of designs (# iterations is set to 25 in this evaluation). But it also requires a dataset of >1000 training points to start out (even if this is limited to low scoring points). That suggests that you 1) must already have a high throughput way of screening fitness values for this task but also 2) want to further optimize a very constrained number of samples. In the case where you are able to generate the low-scoring dataset of size 1000+, does it matter if you find the optimal solution after 25 iterations or after 500 iterations?
* Related to the above point, it would be good to see the performance of the method as a function of the size of the initial training dataset. How well does the method function in a true low-N setting (e.g. < 100 total samples).

Minor Comments
* Section 4.4, points (4) and (5) argue that the length of the sequence is a factor in whether gradient-based methods have a large advantage  over other methods. While I agree this makes sense intuitively, I'm not sure the evidence in the paper supports it. Gradient-based methods still seem to outperform on the DNA tasks, which have a much smaller search space. Additionally, the gap is larger in TFBind10 vs. TFBind8, but not clear exactly how much larger or whether this is actually significant.



**Summary Of The Paper:**

The paper proposes a new method for offline model-based design, which is then applied to biological sequences. The method is able to use a kernel based on pretrained LM features, which allows the method to leverage additional knowledge. In addition, a learning rate adaptation method is proposed for bidirectional learning, with results showing improvement on the chosen benchmarks.

**Summary Of The Review:**

The method is well presented and novel, and would be a reasonable contribution to ICLR. There are some issues with the evaluation and the numbers in the table appear not to be correct (or they are not correctly explained in the paper). If this issue is fixed, the paper would be a good candidate for acceptance, but it must be fixed first.

---

> ### Author Response · Authors · 2022-11-16
> **Response to review questions 1/n**
>
> ## General Reply
>
> We very much appreciate your careful review and the insightful comments and questions. We have modified the paper in light of them, and we believe that it has improved considerably. Below we respond to the comments and explain what changes we have implemented.
>
> ## the scale of the experiments
>
> > Something is odd with the scale of the experiments. When I download the avGFP dataset, the maximum fitness value in the dataset is ~4.12, while the paper reports a value >8. The 4.12 number also appears consistent with the results in Ren et al. 2022. There also seems to be a scale mismatch in the AAV and E4B datasets between this paper and Ren et al. 2022 (in this paper, AAV fitness varies between 0.4 and 0.6, in Ren et al., it varies between -3.5 and 4.5; something similar is true for E4B).
> It would be useful if the authors listed the exact fitness metric for each task and perhaps the maximum value achievable in the dataset. Additionally authors should explain or correct any scale mismatch when comparing with previous work.
> This is the largest issue that must be addressed before publication - I would increase my score if this is addressed satisfactorily.
>
> Thank you for this insightful question which highlights an oversight that we have addressed in a revised version of our paper - we failed to explain exactly how we report performance.
> Following [1, 2], we report the maximum normalized ground truth score $y_n = (y-y_{min})/(y_{max}-y_{min})$. We now state this clearly in Sec. 4.1, and in the table.
> For better illustration, we have changed the title of Table 1 to "Experimental results (maximum normalized ground truth score) for comparison.". Ren et al. [5] do not perform normalization and report the score $y_n = y$.
>
> As for the exact metric for each task, we report the binding activity score for TFBind8(r) and TFBind10(r), the fluorescence level for avGFP, Adeno-associated Viruses viabilities for AAV, the ubiquitination rate for E4B. We believe that the normalized metrics are sufficient for a comparison between methods and also allow us to more readily compare the performance improvements on different datasets. Nevertheless, the raw scores can be valuable, particularly for practitioners who are more familiar with the actual metrics. We have therefore added Table 4 Dataset details in Appendix A.2. This table reports for each dataset the metric that is evaluated, and the minimum and maximum values in the low-scoring (training) and the whole of each dataset.
>
> |  Task  | Metric  | Min of $\mathcal{D}$   | Max of $\mathcal{D}$ | Min of $\mathcal{D}_{entire}$ | Min of $\mathcal{D}_{entire}$
> |:----:|:----:|:----:|:----:|:----:|:----:|
> |D_best | $0.242 $ | $0.248$ | $0.742$ | $0.452$ | $0.224$ |
> TFBind8(r) | binding activity | $0.000$ | ${0.242} $| $0.000$ |  $1.000$
> TFBind10(r) | binding activity | $-1.859$ | ${-0.869} $| $-1.859$ |  $2.129$
> avGFP | fluorescence level | $1.283$ | ${3.553} $| $1.283$ |  $4.123$
> AAV | viruses viability | $-11.176$ | ${-1.399} $| $-11.176$ |  $9.536$
> E4B | ubiquitination rate  | $-3.589$ | ${-0.984} $| $-3.589$ |  $8.998$
>
> ## the incorporation of pretrained language model
>
> > The authors argue that the incorporation of pretrained language model features is a point in favor of their method. It would be interesting to see how the method performs as a function of language model quality. You could use different models from the Elnaggar et al. or the various sizes of the ESM language models and see if improvements in language modeling result in improvements in optimization, which would more directly show that these features have a significant effect on performance.
>
> Thank you very much for your suggestion. This is indeed a very interesting way to gain further insight. We have tested the ProtBERT, ProtAlbert, and ProtBert-BFD models and found that models with better quality generally work better. The publicly available pre-trained DNA models are limited and thus we only perform experiments on the protein tasks. Elnaggar et al. demonstrate that the language model performances follow the ordering: ProtBert-BFD $>$ ProtBert $>$ ProtAlbert. We can see that the performance ranks over the three protein tasks avGFP, AAV, and E4B are the same. We have added these results in Table 5 of Appendix A.4.
>
> |  Pre-trained LM  | avGFP  | AAV  | E4B|
> |:----:|:----:|:----:|:----:|
> ProtAlbert | $3.567\pm {0.456}$ | $0.478\pm {0.004}$ | $0.552\pm {0.023}$
> ProtBert$_{\mathrm{(adopted)}}$ | ${8.084} \pm {0.224}$| $0.501 \pm {0.007}$| ${1.255} \pm {0.029}$
> ProtBert-BFD | $8.240\pm {0.094}$ | $0.549\pm {0.009}$ | $1.880\pm {0.054}$

---

> > ### Author Response · Authors · 2022-11-16
> > **Response to review questions 2/n**
> >
> > ## Hard to follow what metrics
> >
> > > Hard to follow what metrics are being reported in the table - could this be added to the table caption? It’s actually never explicitly said that the reported number is maximum fitness achieved, but I’m assuming it is
> >
> > Following [1, 2], we report the maximum normalized ground truth score $y_n =(y-y_{min})/(y_{max}-y_{min})$. This was stated in the Sec 4.1 evaluation section, but we agree with the reviewer that it should also be in the table caption so that it is not buried in the text, so we have added it in the table caption.
> >
> > ## the problem setting
> >
> > > It would be nice to motivate the problem setting better. I have a hard time understanding when this class of method would be useful. It purports to very quickly optimally sample a set of designs (iterations is set to 25 in this evaluation). But it also requires a dataset of $>1000$ training points to start out (even if this is limited to low scoring points). That suggests that you 1) must already have a high throughput way of screening fitness values for this task but also 2) want to further optimize a very constrained number of samples. In the case where you are able to generate the low-scoring dataset of size 1000+, does it matter if you find the optimal solution after 25 iterations or after 500 iterations?
> >
> > 1. We do not require this optimization to be quick and we set the iteration count to $25$ following [4]. This is enough for convergence and we do not achieve significantly better results with more iterations.
> >
> > 2. We do not require a dataset of $>1000$ training points to start out. Our additional results below show that our approach is successful with as few as $20$ training points, as shown in the table.
> >
> > Generally speaking, in the offline model-based optimization setting, we aim to find some sequences by only leveraging the static dataset and we do not have wet-lab access when performing the algorithms. We can only access wet-lab evaluation when we test different algorithms.
> >
> > The fact that the approach works with $20$ initial values suggests that a practical approach would involve choosing $20$ initial designs (perhaps based on domain expertise, perhaps based on alternative active learning procedures, perhaps randomly chosen). After experiments are conducted for these initial designs, one could then run our proposed method and identify a good design outside the initial set.
> >
> > An alternative practical setting is the scenario where a previous experimental study has already been conducted by other researchers, and we are seeking to perform further experiments and selecting a design based on the available results.
> >
> > |  Dataset size  | $20$  | $40$   | $60$ | $80$ | $100$|
> > |:----:|:----:|:----:|:----:|:----:|:----:|
> > TFBind8(r) | $0.849\pm {0.027}$ | $0.883\pm {0.036}$ | $0.890\pm {0.033}$| $0.911\pm {0.042}$ | $0.923\pm {0.049}$
> > TFBind10(r) | ${0.248} \pm {0.000}$| $0.596 \pm {0.035}$| ${0.602} \pm {0.023}$| $0.616\pm {0.024}$ | $0.632\pm {0.036}$
> >
> > ## a function of the size of the initial training dataset
> >
> > > Related to the above point, it would be good to see the performance of the method as a function of the size of the initial training dataset. How well does the method function in a true low-N setting (e.g. < 100 total samples).
> >
> > Thank you very much for this request. We strongly agree that this would add value to the paper and provide further motivation. We have tested the performance of BIB as a function of dataset size (N= $20$, $40$, $60$, $80$, $100$) in TFBind8(r) and TFBind10(r) since they have exact oracle evaluations. We have added the results in the Appendix A.5 Table 6. We see that performance is already good for N=$20$ for TFBind8(r) and N=$40$ for TFBind10(r) as shown in the table above.

---

> > > ### Author Response · Authors · 2022-11-16
> > > **Response to review questions 3/n**
> > >
> > > ## length of the sequence
> > >
> > > > Section 4.4, points (4) and (5) argue that the length of the sequence is a factor in whether gradient-based methods have a large advantage over other methods. While I agree this makes sense intuitively, I'm not sure the evidence in the paper supports it. Gradient-based methods still seem to outperform on the DNA tasks, which have a much smaller search space. Additionally, the gap is larger in TFBind10 vs. TFBind8, but not clear exactly how much larger or whether this is actually significant.
> > >
> > > It is true that ``Gradient-based methods still seem to outperform on the DNA tasks, which have a much smaller search space.'' What we are trying to stress is that 1) the gradient-based methods have a general advantage on all tasks; and 2) the advantage of gradient-based methods is larger in the long-sequence design. We compute the average ranking of gradient-based methods and non-grad-based methods on TFBind10(r) as $3.5$ and $9.5$, respectively, and the average ranking of gradient-based methods and non-grad-based methods on TFBind8(r) as $5.8$ and $6.8$, respectively. The advantage of gradient-based methods is larger ($9.5-3.5 = 6.0$) in the long-sequence design task than that ($6.8-5.8 = 1.0$) in the short-sequence design task. We have added this analysis to Section 4.4. We consider that the differential ($6.0$ vs. $1.0$) is significant.
> > >
> > > ## Overall
> > >
> > > Thank you again for your instructive feedback on our paper. Please let us know if we have resolved your concern or if you have any further questions.
> > >
> > >     [1] Brandon Trabucco, Xinyang Geng, Aviral Kumar, and Sergey Levine. Design-Bench: benchmarks for data-driven offline model-based optimization. arXiv preprint arXiv:2202.08450, 2022
> > >     [2] Can Chen, Yingxue Zhang, Jie Fu, Xue Liu, and Mark Coates. Bidirectional learning for offline
> > >     infinite-width model-based optimization. Proc. Adv. Neur. Inf. Proc. Syst (NeurIPS), 2022
> > >     [3] Bo Han, Quanming Yao, Xingrui Yu, Gang Niu, Miao Xu, Weihua Hu, Ivor Tsang, and Masashi Sugiyama. Co-teaching: robust training of deep neural networks with extremely noisy labels. Proc.Adv. Neur. Inf. Proc. Syst (NeurIPS), 2018
> > >     [4] Christoffer Norn, Basile IM Wicky, David Juergens, Sirui Liu, David Kim, Doug Tischer, Brian Koepnick, Ivan Anishchenko, David Baker, and Sergey Ovchinnikov. Protein sequence design by conformational landscape optimization. Proceedings of the National Academy of Sciences, 2021.
> > >     [5] Zhizhou Ren, Jiahan Li, Fan Ding, Yuan Zhou, Jianzhu Ma, and Jian Peng. Proximal exploration for model-guided protein sequence design. Proc. Int. Conf. Machine Learning (ICML), 2022.

---

> > > > ### Author Response · Authors · 2022-11-18
> > > > **Kindly Request for Feedback**
> > > >
> > > > Thanks for your detailed review and the questions you raised. We have done the following to address your concerns:
> > > >
> > > > - We have explained and addressed the scale of the experiments;
> > > > - We have conducted experiments to show how the method performs as a function of language model quality;
> > > > - We have added the metrics in the table caption;
> > > > - We have provided a clear motivation of the problem setting;
> > > > - We have tested and reported the performance of our method as a function of dataset size;
> > > > - We have illustrated the relation between the length of the sequence and the advantage of gradient-based methods over non-gradient-based methods;
> > > > - We have updated a revision draft to address all your suggestions and concerns.
> > > >
> > > >
> > > > Thank you again for your instructive feedback on our paper. Please let us know if we have resolved your concerns or if you have any further questions.

---

> > > > > ### Comment · Reviewer_bELc · 2022-11-22
> > > > > **Still confused on the actual numbers**
> > > > >
> > > > > I'm still confused on the normalized ground truth score.
> > > > >
> > > > > ## Reporting Mismatch
> > > > > Looking at avGFP, I see that:
> > > > >
> > > > > $
> > > > > max(D) = 3.553, min(D_{entire}) = 1.283, max(D_{entire}) = 4.123
> > > > > $
> > > > >
> > > > > Plugging this into the formula, I get that the normalized maximum score in the training dataset should be
> > > > >
> > > > > $(max(D) - min(D_{entire})) / (max(D_{entire}) - min(D_{entire})) = (3.553 - 1.283) / (4.123 - 1.283) = 0.7993$.
> > > > >
> > > > > However, the number reported for $D(best)$ is 0.742. Perhaps I have misunderstood some part of this computation?
> > > > >
> > > > > ## Unrealistic Results
> > > > > The optimized result reported has a normalized fitness score of 8.084. This implies an actual log-fluorescence value of 24.24. This value is, to say the least, unrealistic, as it implies the model has discovered a protein that is 10^20 times more fluorescent than the most fluorescent GFP in the dataset (I believe the highest values in this experiment were already reaching the limit of the instrumentation to measure).
> > > > >
> > > > > This suggests the method is not optimizing the underlying landscape, but is instead exploiting the function predictor to find adversarial examples.

---

> > > > > > ### Author Response · Authors · 2022-11-24
> > > > > > **Response to review questions**
> > > > > >
> > > > > > Thanks for your timely feedback!
> > > > > >
> > > > > > ## Reporting Mismatch
> > > > > >
> > > > > > Thank you for checking this. We were not sufficiently clear about our procedure; we explained that we followed the process in previous work in the offline model optimization literature [1], but we should have repeated the details of that procedure.
> > > > > >
> > > > > > When computing the normalized maximum score in the training dataset, we relabel the whole training set using the trained oracle, to remove train-test discrepancy between the oracle and the ground-truth. This follows the procedure in Appendix A.3. Benchmarking Details of [1]. We adopted it to be consistent with previous work. After the replacement, we then get the maximum score of the training set as $max(\mathcal{D}) =3.390$ instead of $max(\mathcal{D}) =3.553$ (the original raw score).
> > > > > >
> > > > > > This leads to a normalized score as $(3.390 - 1.283)/(4.123 - 1.283) = 0.742$. Similarly, AAV has $max(\mathcal{D}) =-1.814$ and E4B has $max(\mathcal{D}) =-0.770$, which yields the normalized maximum score as $0.452$ and $0.224$ respectively.
> > > > > >
> > > > > > In light of the points you raise, and the confusion it has caused, we agree that it is better to depart from the procedure in [1] and instead normalize using the raw max and min scores in the dataset.
> > > > > >
> > > > > >
> > > > > > ## Unrealistic Results
> > > > > >
> > > > > > Thanks for pointing this out. It is a concerning result. The log-fluorescence value reported by the oracle is $24.24$, five times larger than the most fluorescent GFP in the dataset.
> > > > > >
> > > > > > The oracle is clearly inaccurate, but it is the best oracle we can find for comparing different methods. In order to select an oracle, we identified which model could achieve the highest Pearson correlation coefficient for the validation set.
> > > > > > Specifically, we randomly split the entire dataset into a training set and a validation set. We trained the oracle on the training set with different seeds and architectures, and finally selected the one that achieved the highest Pearson correlation coefficient for the validation set ($0.746$). This value suggests that the oracle does achieve a relatively high correlation with the values in the dataset. The absolute error for a given validation point can be quite large, and the model is not restricted to a reasonable range of fluorescence values.
> > > > > >
> > > > > > We experimented with other trained oracles but we found that these oracles are even poorer, and yield indistinguishable results across all methods. This observation is also reported in Appendix D in [2].
> > > > > >
> > > > > > For each of the other four tasks, the score of the best design is very close to the maximum score of $\mathcal{D}_{entire}$. The Pearson correlation coefficient for each of the oracles is closer to one and the absolute errors on the validation set much smaller. This suggests that, for the other tasks, performance evaluation using the oracle is likely to be reliable indicator of the capabilities of the methods.
> > > > > >
> > > > > > If we remove avGFP from consideration, our method is still the best performing method in terms of rank mean and rank median, as shown below.
> > > > > >
> > > > > > |  Method  | TFBind8(r)  | TFBind10(r)   | AAV | E4B | Rank Mean | Rank Median
> > > > > > |:----:|:----:|:----:|:----:|:----:|:----:|:----:|
> > > > > > |D_best | $0.242 $ | $0.248$ | $0.452$ | $0.224$ | |  |
> > > > > > |BO-qEI | $0.940 \pm {0.032}$ | $0.595 \pm {0.028}$ | $\textbf{0.591} \pm \textbf{0.002}$|  $0.436 \pm {0.004}$ | $5.8/12$ | $5.5/12$|
> > > > > > | CMA-ES | $0.930\pm {0.034}$ | $0.617\pm {0.031}$ | $0.470\pm {0.006}$| $0.748\pm {0.009}$ | $6.8/12$ | $6.5/12$
> > > > > > AdaLead | ${0.941} \pm { {0.032}}$ | $0.602\pm {0.028}$ | $0.581\pm {0.002}$| $0.433\pm {0.003}$ | $5.8/12$ | $5.5/12$
> > > > > > |CbAS | $0.878\pm {0.049}$ | $0.610\pm {0.035}$ | $0.543\pm {0.002}$| $0.349\pm {0.003}$ | $8.3/12$ | $9.5/12$
> > > > > > |PEX | $0.924\pm {0.041}$ | $0.612\pm {0.026}$ | ${0.588} \pm {0.002}$ | $0.397 \pm {0.004}$ | $6.8/12$ | $8.0/12$
> > > > > > |GENH | $0.323\pm {0.000}$ | $0.448\pm {0.000}$ | $0.452\pm {0.000}$| $0.228\pm {0.000}$ | $11.5/12$ | $11.5/12$
> > > > > > |Grad | ${0.941} \pm {0.026}$ | $0.630\pm {0.029}$ | $0.463\pm {0.005}$| ${1.219} \pm {0.061}$ | $5.0/12$ | $4.0/12$
> > > > > > |COMs | $0.921\pm {0.039}$ | {$0.637\pm {0.065}$} | $0.511\pm {0.005}$| $0.829\pm {0.026}$ | $5.5/12$ | $5.0/12$
> > > > > > |ROMA | $0.926\pm {0.032}$ | $0.634\pm {0.061}$ | $0.471 \pm {0.005}$| $1.198 \pm {0.042}$ | $5.5/12$ | $5.5/12$
> > > > > > |NEMO | $0.930\pm {0.038}$ | $0.632\pm {0.024}$ | $0.505\pm {0.005}$| $1.036\pm {0.046}$ | $5.0/12$ | $5.0/12$
> > > > > > |BDI | $0.823\pm {0.000}$ | $\textbf{0.678}\pm \textbf{0.000}$ | $0.452\pm {0.000}$| $0.224\pm {0.000}$ | $8.8/12$ | $11.0/12$
> > > > > > |BIB$_{\mathrm{(ours)}}$ | $\textbf{0.952} \pm \textbf{0.033}$ | ${0.639} \pm {0.032}$ | $0.501 \pm {0.007}$| $\textbf{1.255} \pm \textbf{0.029}$ | $2.8/12$ | $1.5/12$
> > > > > >
> > > > > >     [1] Brandon Trabucco, Aviral Kumar, Xinyang Geng, and Sergey Levine. Conservative objective models
> > > > > >     for effective offline model-based optimization. ICML 2021.
> > > > > >     [2] Brandon Trabucco, Xinyang Geng, Aviral Kumar, and Sergey Levine. Design-Bench: benchmarks for data-driven offline model-based optimization. ICML 2022.

---

### Official Review · Reviewer_q1Be · 2022-10-24

**Confidence:** 4
**Correctness:** 2
**Technical Novelty And Significance:** 2
**Empirical Novelty And Significance:** 2
**Recommendation:** 6

**Clarity, Quality, Novelty And Reproducibility:**

The paper combines a couple of orthogonal ideas to extend a recent paper on bidirectional model-based optimization (Chen 2022). Each of these ideas is clever, but has appeared in other ML contexts.

I found the exposition of the bidirectional learning method very confusing. It can only be understood by reading the original paper.

I have some large reservations regarding the quality of the analysis of the experimental results (see below).




**Strength And Weaknesses:**

=Strengths=
Paper contributes to an existing active thread on offline model-based optimization, which is an interesting and important area.
Paper tackles offline biological sequence design, which is a practical application with large potential impact.
Paper combines a number of modern techniques, such as neural network linearization, gradient-based hyper-parameter tuning using bilevel optimization, etc.

=Weakness=
I found the exposition of the bidirectional learning method extremely confusing.
See my full review below regarding my hesitations around the setup of the experiments.

**Summary Of The Paper:**

A recent 'bidirectional learning' approach was proposed for offline model-based optimization. This paper extends it with a few tweaks: instead of using an NTK kernel, use the linearization of a pretrained foundation model, and tune optimizer hyper-parameters automatically by gradient descent using an auxiliary model to provide pseudo-labels for held-out data. The experiments focus on protein/dna design tasks and consider a number of methods from recent work.

**Summary Of The Review:**

A have some important reservations about the paper below.

==Parametrization of the oracle vs. the regression model==
For datasets where the entire search space hasn't been measured (i.e., everything other than the tfbind datasets), you define the ground truth f(x) by fitting a regression model (the 'oracle') on the entire dataset, such that this model can be queried on any x. I'm very concerned that the same overall model architecture is used for the oracle, the methods' function approximators f_theta, and f_aux:
*Oracle: "The oracle passes the average of the residue embeddings from the pre-trained Prot-T5 (Elnaggar et al., 2021) into a linear layer and then fits the dataset. The following two task oracles take the same form."
*Function approximator: "In this paper, we adopt the pre-trained DNABERT (Ji et al., 2021) and Prot-BERT (Elnaggar et al., 2021) models, and compute the average of token hidden embeddings as the extracted feature, which is fed into the linear layer to build the proxy. … Following (Dukler et al., 2022), we can also only linearize the last layer of the network for simplicity."
*Auxiliary model f_aux: ""We implement the auxiliary model as a linear layer with the feature from the pre-trained LM?"

This setup will unfairly favor methods that don't rely on a function approximator (i.e., it will penalize methods like CMA-ES), since the function approximator will have low approximation error for fitting the oracle, since they're from the same function class. It will additionally favor methods that use f_aux and model linearization, since these closely mirror the structure of the oracle. This is not a realistic setup. Model-misspecification will always be present when trying to model real experimental data with a neural network.

An alternative approach would avoid the oracle by evaluating optimization methods' ability to rank a held-out set of examples from the dataset (ranking them by the methods' associated acquisition function). I would have preferred to see some experiments seeing the second approach, since it removes the above issues.

==Presentation of the paper's method==
I found section 3.1 extremely confusing. There should be a background section that explains the bidirectional learning approach fully. The paper should be self-contained. Right now, readers will not be able to understand the explanation of how it works. The datasets Xh and Xl are never defined. I was completely unclear to me where yh comes from. I assumed that Xh were prospective designs and yh was unobserved. It was also never explained how the two models are combined into an acquisition function.

==Complex model parameterization==
I work in this exact research field. In my experience, simple linear models are often quite competitive when modeling these datasets. I would have appreciated some demonstration that the particular modeling choice you employ is sensible.

==Presentation of experimental results==
It was very difficult for me to understand if any of the difference between methods' performances in Table 1 are statistically significant. As far as I can tell, all of these optimization methods are randomized. How much variance in performance is due to this randomness? I would have appreciated some error bars that convey this variance.

==Role/Importance of f_aux==
sec 3.2 is something you could do for basically ML system: just train two models and use M1 to provide aux labels for hyper tuning. for this to work, the aux model needs to be accurate, but if you're able to fit a good aux model, then isn't the overall problem easy?

==Additional issues==

"Both DNA tasks have exact oracles for ground-truth evaluation."
This isn't quite true. The datasets measure the entire search space, but these measurements come from a noisy wet-lab experiment. We never observe the true noiseless value.

Fig 2 was not compelling to me. Where in sec 3.2 does gamma depend on T? Are you simply plotting the trajectory of the learned scalar gamma over the course of hyper-tuning. The shape of this trajectory should not be important.

---

> ### Author Response · Authors · 2022-11-16
> **Response to review questions 1/n**
>
> ## General Reply
>
> Many thanks for your valuable and constructive comments. These will help us to clarify, correct, and improve the materials in this paper! We will carefully revise the paper according to your comments as explained below.
>
> ## Parametrization of the oracle vs. the regression model
>
> > ==Parametrization of the oracle vs. the regression model== you define the ground truth f(x) by fitting a regression model (the 'oracle') on the entire dataset, such that this model can be queried on any x. I'm very concerned that the same overall model architecture is used for the oracle, the methods' function approximators ftheta, and faux: *Oracle: This setup will unfairly favor methods that dont rely on a function approximator (i.e., it will penalize methods like CMA-ES), since the function approximator will have low approximation error for fitting the oracle, since theyre from the same function class. It will additionally favor methods that use faux and model linearization, since these closely mirror the structure of the oracle. This is not a realistic setup. Model-misspecification will always be present when trying to model real experimental data with a neural network. An alternative approach would avoid the oracle by evaluating optimization methods ability to rank a held-out set of examples from the dataset (ranking them by the methods associated acquisition function). I would have preferred to see some experiments seeing the second approach, since it removes the above issues.
>
>
> ### Potential correlation between the oracle and the proxy.
>
> We agree with your concern regarding the potential bias. We note that the oracle approach is standard throughout the offline optimization literature. The difficulty is that we do need a way to evaluate proposed designs that are not in the original test set. We attempted to avoid substantial bias by choosing considerably different models for the proxy and the oracle. The parameterization of the oracle is different from that of the regression model from two aspects: 1) model architecture; 2) pre-trained information source. First, the oracle adopts the Prot-T5 model which consists of an encoder and a decoder, while the regression model adopts the Prot-BERT model which only has an encoder. Second, Prot-T5 is trained on the BFD and UniRef100 datasets and ProtBert is trained on the UniRef50 dataset.  We choose the Prot-T5 model as the oracle because this is the state-of-the-art protein LM to extract features and recent work [1] has demonstrated its effectiveness. In order to test how related the Prot-T5 (oracle)/Prot-BERT(proxy) models are, we trained them on a sampled training dataset and compared the test predictions of the testing set. By evaluating the Pearson correlation coefficient (PCC) between the two prediction errors PCC(ProtT5 predictions - test labels, ProtBERT predictions -test labels), we obtain $-0.0053$ on avGFP, $-0.0005$ on AAV, and $-0.0062$ on E4B. These results suggest that the two models are not strongly related in terms of the predictions they form. We have added the above explanations in Appendix A.2.
>
> ### Unfairly favor methods
>
> All methods rely on the function approximator. In CMA-ES, the proposed sequences will be labeled by the function approximator and these labeled instances are further used to improve the covariance matrix adaptation. One could perhaps argue that some methods are more reliant on the accuracy of the proxy function approximator than others, but it is dangerous to state this as a fact without careful investigation.
>
>
> ### The problem with the second approach
>
> The alternative approach you propose cannot be used to validly compare the different optimization methods. The key aim of the problem setting is to propose new sequences that are out-of-distribution (with respect to the training set). The goal is to find one or more good designs; in order to do this, a model does not have to provide good predictions for the entire test set. So it becomes a challenge to define a suitable test set for comparing the final models.
>
>
> Further, many methods, including BO-qEI, CMA-ES, and PEX, have multiple rounds and we cannot use a simple acquisition function to characterize these methods without considering the strategy in the multi-round procedure. Since you mention CMA-ES, we will use CMA-ES to illustrate this: in an iteration, CMA-ES samples some high-scoring sequences according to the covariance matrix and labels these sequences using the function approximator; then the top-scoring sequences are used to update the covariance matrix. Even the gradient-based methods repeatedly adjust the model during the acquisition process. At each step, it is desirable for the model to be locally accurate, but it is not necessary for it to be accurate over the entire design space. The relative performance of predictions for a held-out test set for these local predictive models is not indicative of the capabilities of the optimization techniques themselves.

---

> > ### Author Response · Authors · 2022-11-16
> > **Response to review questions 2/n**
> >
> > ## Presentation of the paper's method
> >
> > > ==Presentation of the paper's method== I found section 3.1 extremely confusing. There should be a background section that explains the bidirectional learning approach fully. The paper should be self-contained. Right now, readers will not be able to understand the explanation of how it works. The datasets $X_h$ and $X_l$ are never defined. I was completely unclear to me where $y_h$ comes from. I assumed that Xh were prospective designs and $y_h$ was unobserved. It was also never explained how the two models are combined into an acquisition function.
> >
> > We apologize for the lack of clarity and have modified the paper to address this concern. We have added a background part with a detailed figure in Section 2.4 to introduce bidirectional learning according to your suggestion. Although in the previous version of the paper,  we did define ($X_h$, $y_h$) and ($X_l$, $y_l$) in Section 3.1, we did not make it very clear and there was not enough detail regarding bidirectional learning. We have now defined these more clearly in the background Section 2.4.
> >
> > Specifically, the data $X_h$ consists of the high-scoring sequences we aim to find and the dataset ($X_l$, $y_l$) is the static dataset we have collected. $y_h$ is a predefined large score. Bidirectional learning works as follows. The forward mapping leverages the static dataset to predict the scores of the high-scoring sequences, and the backward mapping leverages the high-scoring sequences to predict the scores of the static dataset. By minimizing the prediction losses, we aim to find suitable high-scoring designs.
> >
> > ## Complex model parameterization
> >
> > > {==Complex model parameterization== I work in this exact research field. In my experience, simple linear models are often quite competitive when modeling these datasets. I would have appreciated some demonstration that the particular modeling choice you employ is sensible.
> >
> > Thank you for your good suggestion!
> > Could you illustrate what kind of linear models you are referring to? We would be pleased to compare to an appropriate model, but are concerned that our choice of model would not match what you had in mind.
> >
> > Our current model can actually be viewed as a simple linear model with a (complicated) pre-trained feature extractor. We do not fine-tune the pre-trained model. It simply extracts features and passes these to a linear layer, forming a regression model. So perhaps the linear models you have in mind are similar to what we end up using, just with a different feature extraction procedure. As demonstrated in [1, 2], the modeling technique is effective for protein sequence property prediction. Recent work [3] on protein sequence design also adopts the linear model with a pre-trained feature extractor.
> >
> > If you are referring to a CNN model on the one-hot encoding as a linear model, this could potentially provide a strong baseline if there are many data points for training. When there is limited data, as is often the case in the drug discovery/protein design area, the adoption of the pre-trained LM is usually a better approach. This observation is supported by the experimental results in [5].

---

> > > ### Author Response · Authors · 2022-11-16
> > > **Response to review questions 3/n**
> > >
> > > ## Presentation of experimental results
> > >
> > > > ==Presentation of experimental results== It was very difficult for me to understand if any of the difference between methods' performances in Table 1 are statistically significant. As far as I can tell, all of these optimization methods are randomized. How much variance in performance is due to this randomness? I would have appreciated some error bars that convey this variance.
> > >
> > > Yes, the performances between methods are significant. We run statistical significance tests (Wilcoxon signed-rank test) and compute the p-values between the best method and the second-best method are $0.0003$ on avGFP, $0.0002$ on AAV, $0.005$ on E4B, $0.348$ on TFBind8(r) and $0.0006$ on TFBind10(r). This demonstrates our method is significantly better than other methods in a majority of tasks. The search space of DNA sequence design is limited and thus can not reveal the significant difference between comparison methods.
> > >
> > > The mean and standard deviation of our method BIB over 16 runs are $0.952 \pm 0.033$ on TFBind8(r), $0.639 \pm 0.032$ on TFBind10(r), $8.084 \pm 0.224$ on avGFP, $0.501 \pm 0.007$ on AAV and $1.255 \pm 0.029$ on E4B. The detailed results are in the following table and we have modified the table in the main paper. As we can see, the error bars are relatively small, which demonstrates the robustness of this setting.
> > >
> > > |  Method  | TFBind8(r)  | TFBind10(r)   | avGFP | AAV | E4B | Rank Mean | Rank Median
> > > |:----:|:----:|:----:|:----:|:----:|:----:|:----:|:----:|
> > > |D_{best} | $0.242 $ | $0.248$ | $0.742$ | $0.452$ | $0.224$ | |  |
> > > |BO-qEI | $0.940 \pm {0.032}$ | $0.595 \pm {0.028}$ | $1.700 \pm {0.020}$|  $\textbf{0.591} \pm \textbf{0.002}$|  $0.436 \pm {0.004}$ | $6.0/12$ | $7.0/12$|
> > > | CMA-ES | $0.930\pm {0.034}$ | $0.617\pm {0.031}$ | $5.488\pm {0.056}$| $0.470\pm {0.006}$| $0.748\pm {0.009}$ | $6.0/12$ | $6.0/12$
> > > AdaLead | ${0.941} \pm { {0.032}}$ | $0.602\pm {0.028}$ | $1.611\pm {0.009}$| $0.581\pm {0.002}$| $0.433\pm {0.003}$ | $6.2/12$ | $8.0/12$
> > > |CbAS | $0.878\pm {0.049}$ | $0.610\pm {0.035}$ | $1.371\pm {0.016}$| $0.543\pm {0.002}$| $0.349\pm {0.003}$ | $8.6/12$ | $10.0/12$
> > > |PEX | $0.924\pm {0.041}$ | $0.612\pm {0.026}$ | $1.546\pm {0.019}$| ${0.588} \pm {0.002}$ | $0.397 \pm {0.004}$ | $7.2/12$ | $8.0/12$
> > > |GENH | $0.323\pm {0.000}$ | $0.448\pm {0.000}$ | $0.835\pm {0.000}$| $0.452\pm {0.000}$| $0.228\pm {0.000}$ | $11.4/12$ | $11.0/12$
> > > |Grad | ${0.941} \pm {0.026}$ | $0.630\pm {0.029}$ | $4.869\pm {0.042}$| $0.463\pm {0.005}$| ${1.219} \pm {0.061}$ | $4.8/12$ | $4.0/12$
> > > |COMs | $0.921\pm {0.039}$ | $0.637\pm {0.065}$ | $3.873\pm {0.080}$| $0.511\pm {0.005}$| $0.829\pm {0.026}$ | $5.6/12$ | $5.0/12$
> > > |ROMA | $0.926\pm {0.032}$ | $0.634\pm {0.061}$ | ${5.621} \pm {0.143}$| $0.471 \pm {0.005}$| $1.198 \pm {0.042}$ | $4.8/12$ | $4.0/12$
> > > |NEMO | $0.930\pm {0.038}$ | $0.632\pm {0.024}$ | $4.624\pm {0.087}$| $0.505\pm {0.005}$| $1.036\pm {0.046}$ | $5.0/12$ | $5.0/12$
> > > |BDI | $0.823\pm {0.000}$ | $\textbf{0.678}\pm \textbf{0.000}$ | $0.742\pm {0.000}$| $0.452\pm {0.000}$| $0.224\pm {0.000}$ | $9.4/12$ | $11.0/12$
> > > |BIB | $\textbf{0.952} \pm \textbf{0.033}$ | ${0.639} \pm {0.032}$ | $\textbf{8.084} \pm \textbf{0.224}$| $0.501 \pm {0.007}$| $\textbf{1.255} \pm \textbf{0.029}$ | $2.4/12$ | $1.0/12$
> > >
> > > ## Role/Importance of faux
> > >
> > > > ==Role/Importance of faux== sec 3.2 is something you could do for basically ML system: just train two models and use M1 to provide aux labels for hyper tuning. for this to work, the aux model needs to be accurate, but if you're able to fit a good aux model, then isn't the overall problem easy?
> > >
> > > Thank you for this insightful question. Our idea is motivated by co-teaching [4], where two models leverage each other's views to filter noisy data. In co-teaching, the auxiliary model evaluates the loss of every sample and feeds the small loss samples (more likely to be clean samples) to the model. It thus provides a weak supervision signal. Importantly it is observed that even a very weak auxiliary model can improve the performance of a much superior predictive model when used in this way. By contrast, a direct combination of predictions can lead to worse performance. In [4], we can view the decision as to whether the sample is chosen or not as a ``hyper-parameter'', and the aux model tunes this hyper-parameter to provide weak supervision signals.
> > >
> > > In our case, the aux model evaluates the more traditional model hyperparameters, including the trade-off parameter and the learning rate. The model thus provides a weak supervision signal to the bidirectional learning process.
> > > The key insight is that the aux model does not need to be very accurate since it only needs to provide a weak supervision signal to update the hyperparameters instead of targetting the main optimization objective.  We have demonstrated this point in Sec 4.5 where experimental results show that the aux model cannot serve to optimize the main objective directly.

---

> > > > ### Author Response · Authors · 2022-11-16
> > > > **Response to review questions 4/n**
> > > >
> > > > ## Minor issues
> > > >
> > > > > "Both DNA tasks have exact oracles for ground-truth evaluation." This isn't quite true. The datasets measure the entire search space, but these measurements come from a noisy wet-lab experiment. We never observe the true noiseless value.
> > > >
> > > >
> > > > Thanks for your clarification. We have refined the sentence in Appendix A.2 of the latest version as follows: ``Both DNA tasks measure the entire search space and we adopt these measurements as the approximate ground-truth evaluation.''
> > > >
> > > > > Fig 2 was not compelling to me. Where in sec 3.2 does gamma depend on T? Are you simply plotting the trajectory of the learned scalar $\gamma$ over the course of hyper-tuning. The shape of this trajectory should not be important.
> > > >
> > > > $\gamma$ does not depend on T in Section 3.2. As we illustrate in Algorithm 1 Line 3 of the revised version, we update gamma in every sequence optimization iteration. We plot the trajectory of the learned scalar gamma over the course of hyper-tuning and this trajectory is informative and important because it can better illustrate the effectiveness of the adaptive-$\gamma$ module and potentially give us some biological insights.
> > > >
> > > > $\gamma$ controls a trade-off between exploration and constraint. The aux model provides weak supervision signals to update the hyperparameter. As we illustrate in Section 4.4, we find that the $\gamma$ values of TFBind8(r) and avGFP generally increase at first. This means that BIB reduces the impact of the constraint to encourage a more aggressive search for a high target value during the initial phase. Then $\gamma$ of TFBind8(r) continues to increase while the $\gamma$ of avGFP decreases. We conjecture that the difference is caused by the sequence length. Small mutations of a biological sequence are enough to yield a good candidate. For the length-$239$ protein in avGFP, dramatic mutations 1) are not necessary and 2) can easily lead to out-of-distribution points.
> > > > The weak supervision signal from the auxiliary model therefore
> > > > encourages a tighter constraint towards the static dataset. By contrast, the DNA sequence is relatively short and a more widespread search of the sequence space can yield better results. To investigate this conjecture, we further visualize the trend of E4B. E4B also has long sequences ($102$) and we can observe its similar
> > > > first-increase-then-decrease trend, although it is not as pronounced.
> > > >
> > > > ## Overall
> > > >
> > > > Does the above reply address your concerns? Thank you again for your instructive review and feedback. We very much appreciate your careful review of the paper and look forward to further exchange with you during the rebuttal phase.
> > > >
> > > >     [1] Ahmed et al. ProtTrans: towards cracking the language of lifes code through self-supervised deep learning and high performance computing. IEEE Transactions on Pattern Analysis and Machine Intelligence, 2020.
> > > >     [2] Ji et al. DNABERT: pre-trained bidirectional encoder representations from transformers model for DNA-language in genome. Bioinformatics, 2021.
> > > >     [3] Zhizhou Ren, Jiahan Li, Fan Ding, Yuan Zhou, Jianzhu Ma, and Jian Peng. Proximal exploration for model-guided protein sequence design. Proc. Int. Conf. Machine Learning (ICML), 2022.
> > > >     [4] Bo Han, Quanming Yao, Xingrui Yu, Gang Niu, Miao Xu, Weihua Hu, Ivor Tsang, and Masashi Sugiyama. Co-teaching: robust training of deep neural networks with extremely noisy labels. Proc. Adv. Neur. Inf. Proc. Syst (NeurIPS), 2018.
> > > >     [5] Dallago C, Mou J, Johnston K E, et al. FLIP: Benchmark tasks in fitness landscape inference for proteins[J]. bioRxiv, 2021.

---

> > > > > ### Author Response · Authors · 2022-11-18
> > > > > **Kindly Request for Feedback**
> > > > >
> > > > > Thank you for your detailed review and your constructive feedback. We tried to address all your listed concerns above, mainly including:
> > > > >
> > > > > - We have explained the difference between the oracle and the proxy. We have conducted experiments to verify that there is limited correlation between their errors with respect to the test data;
> > > > > - We have clarified that all methods rely on a function approximator and thus there is no evidence that this evaluation protocol will unfairly favor gradient-based methods;
> > > > > - We have illustrated the problem with the second evaluation approach proposed by the reviewer;
> > > > > - We have added a background discussion with a detailed figure in Section 2.4 to make the presentation of the paper's method clearer;
> > > > > - We have demonstrated why the particular modeling choice is sensible;
> > > > > - We have run statistical significance tests (Wilcoxon signed-rank test) and reported the error bars to make our experimental results more solid;
> > > > > - We have illustrated the role/importance of the auxiliary model. It provides weak supervision signals to update the hyperparameters;
> > > > > - We have refined the descriptions of the DNA tasks and explained the importance of the shape of the trajectory;
> > > > > - We have submitted a revision draft to address all your concerns.
> > > > >
> > > > > Could you please further elaborate if anything is still not clear? Thank you and looking forward to your feedback.

---

### Decision · Program_Chairs · 2023-01-20

**Decision:**

Reject

**Justification For Why Not Higher Score:**

* Contributions not well supported.
* Differences with respect to closely-related work not sufficient to properly assess the contributions.
* Clarity/formal presentations.

**Justification For Why Not Lower Score:**

N/A

**Metareview: Summary, Strengths And Weaknesses:**

The reviewers and meta reviewer all carefully checked and discussed the rebuttal. They thank the authors for their response and their efforts during the rebuttal phase.
The response helped resolve some concerns (e.g., further experimental details and some analysis of the ranking order of the surrogate models).

The reviewers and meta reviewer all acknowledge that the submission tackles an important problem, namely the (model-based) optimization of biological sequences.

This being said, the submission still suffers from important flaws that would warrant further consolidations/investigations. In particular:

(i) The contributions 2 and 3 (“adaptive-gamma” and “adaptive-eta”) do not seem to relate at all to the vast existing literature about bi-level optimization addressing similar problems, e.g., (Maclaurin et al., 2015; Luketina et al., 2016; Pedregosa, 2016; Franceschi et al., 2018; Lorraine et al., 2020) to name a few.  As a result, it makes it difficult to assess the substance of those two contributions. For example, “... the inner level solution can be approximated via a gradient descent step with a learning rate…” seems to correspond to the method of Luketina et al., 2016. Similarly, the extension to the optimization of the learning rate (“adaptive-eta”) has strong connections with work like Donini et al., 2020 (MARTHE: Scheduling the Learning Rate Via Online Hypergradients) & references therein.
To make contributions 2 and 3 actual contributions, precise discussions of the related work and extensive comparisons should be carried out (especially because the claims in the submission are strong, e.g., “...the first learning rate adaptation module for offline model-based optimization…”).

(ii) Regarding contribution 1, it would be important to more clearly state what is inherited from Chen et al. 2022 and what is novel in the submission. Several sections appear to be heavily inspired by Chen et al. 2022 (e.g., the ablation study about forward vs. backward).

(iii) The exposition of the methodology needs to be clarified and to be made more formal (along the lines of the exposition in Chen et al. 2022). Also some expressions seem to be mistaken (e.g., lack of second-order derivative in equation (21)). Analogously, the equation (6) introduces non-differentiability (argmax), whose implications are not discussed.

Because of the extremely competitive landscape of the submissions this year, the paper remains under the cut and has ultimately not been selected for acceptance.
We are convinced that the suggestions above will help strengthen the paper for a future resubmission, which the reviewers and meta reviewer all encourage.

As a final suggestion, with the paper being centered around protein design, there are differences between optimizing a synthetic neural-network landscape and a true protein fitness landscape. In the current state, we have little evidence that the methods being currently developed will succeed on both the former and the latter. Since the former is an arguably very difficult goal, it would be better to frame/tone down the narrative of the paper and set expectations accordingly, e.g., saying that the paper primarily focuses on a proxy, synthetic problem.


**Summary Of Ac-Reviewer Meeting:**

The meta reviewer and three (out of the three) reviewers met on 12/9.

In the meeting, we quickly converged to a reject decision by collectively discussing the three points listed in the meta review, namely:

(i) Contributions 2 and 3 in the paper are not studied/backed up enough.

(ii) The differences with respect to (Chen et al., 2022) need to be clarified to better assess the contributions of the submission.

(iii) Clarity/formal presentation concerns.

Finally, the attendees of the meeting discussed at length other suggestions to improve the manuscript. In particular (see last paragraph of the meta review), how to best frame the narrative with respect to the protein design aspect of the paper.